

# Using StorAge Selection functions to quantify ecohydrological controls on the time-variant age of evapotranspiration, soil water, and recharge

Aaron A. Smith[1], Doerthe Tetzlaff[1,2,3], Chris Soulsby[1]

[1]Northern Rivers Institute, School of Geosciences, University of Aberdeen, UK
[2]Humboldt University Berlin, Berlin, Germany
[3]IGB Leibniz Institute of Freshwater Ecology and Inland Fisheries Berlin, Berlin, Germany

*Correspondence to*: Aaron A. Smith (aaron.smith@abdn.ac.uk)

**Abstract.** Quantifying ecohydrological controls on soil water availability is essential to understand temporal variations in
catchment storage. Soil water is subject to numerous time-variable fluxes (evaporation, root-uptake, and recharge), each with
different water ages which in turn affect the age of water in storage. Here, we adapt StorAge Selection (SAS) function theory
to investigate water flow in soils and identify soil evaporation and root-water uptake sources from depth. We use this to
quantify the effects of soil-vegetation interactions on the inter-relationships between water fluxes, storage, and age. The novel
modification of the SAS function framework is tested against empirical data from two contrasting soil-vegetation units in the
Scottish Highlands; these are characterised by significant preferential flow, transporting younger water through the soil during
high soil moisture conditions. Dominant young water fluxes, along with relatively low rainfall intensities, explain relatively
stable soil water ages through time and with depth. Soil evaporation sources were more time-invariant with high preference
for near-surface water, independent of soil moisture conditions, and resulting in soil evaporation water ages similar to near-
surface soil waters (mean age: 50 – 65 days). Sources of root-water uptake were more variable: preferential near-surface water
uptake occurred in wet conditions, with a deeper root-uptake source during dry soil conditions, which resulted in more variable
water ages of transpiration (mean age: 56 – 79 days). The simple model structure provides a parsimonious means of
constraining the water age of multiple fluxes from the upper part of the critical zone during time-varying conditions improving
our understanding of vegetation influences on catchment scale water fluxes.

## 1 Introduction

Recent studies in vegetation-soil interactions have suggested that soil waters contributing to recharge and streamflow, and soil
waters available for vegetation root-uptake are effectively de-coupled systems, giving rise to the "two-water worlds
hypothesis" (McDonnell, 2014; Evaristo et al., 2015; Berry et al., 2017). Water stored in the soil profile is dynamic and
continuously subject to time-variable fluxes (evaporation, root-uptake, recharge), each with different water ages which in turn
affect the age of water in storage. Whilst isotopic data in some regions has infer a clear separation of plant water from soil



water, this is not apparent in all cases (Ellsworth and Williams, 2007; Vargas et al., 2017), and the source of water for vegetation and their uptake processes needs further methodological development and testing (Rothfuss and Javaux, 2017). Quantifying ecohydrologic separation is essential for understanding the influence of vegetation and critical zone processes on catchment scale water fluxes. Within the critical zone, soil water movement is highly complex with heterogeneous soil

properties and pressure gradients creating highly preferential flow and depth-dependent soil sources of evapotranspiration (*ET*), which are generally simplified to a net flux in large-catchment scale modelling approaches (Zhao et al., 2013). The characterization of the water ages of evaporation and root-uptake as well as their fluxes from discrete depths have been subject to limited investigation. The identification of root-water uptake ages and fluxes is often difficult; this is due to the often unknown root densities and soil moisture distributions that influence the spatial location of preferential root-uptake volumes

(Brantley et al., 2017). Isotopes have been shown to be a useful tool to identify root-uptake source through mass-balance of soil isotopic compositions and xylem water samples (Ogle et al., 2014; Geris et al., 2017; Sprenger et al., 2017b). Similar to root-uptake, evaporation fluxes and age from depth are difficult to quantify due to limitations in measured energy fluxes (Xiao et al., 2011, 2012) which may be used to infer evaporation sources during dry and wet conditions (Sakai et al., 2011). Studies have previously inferred the age of evaporative water using simple flux tracking, and suggest significantly younger removal

via evaporation relative to stream water at the catchment outlet (Soulsby et al., 2016).

StorAge Selection functions provide a parsimonious method of incorporating multiple processes while identifying the water ages of different fluxes from storage (Botter et al., 2011). These functions provide insight into the interrelationship of multiple fluxes and the preferential removal of old or young water. The framework for using time-variant distributions to temporally differentiate water ages from storages has been defined by the "master equation" (Botter et al., 2011), identifying how water

preferentially moves through storage. The majority of time-variant approaches to assessing water age have focused on catchment-scales, however, transit and residence times have also been inferred in lysimeter studies through the use of tracer injections and breakthrough curves (Rinaldo et al., 2011; Harman and Kim, 2014; Benettin et al., 2015; Queloz et al., 2015; Kim et al., 2016). This has led to the identification of time-variant changes of transit time in different soils, directly related to moisture content (Ali et al., 2014; Tetzlaff et al., 2014; Sprenger et al., 2016; Pangle et al., 2017). There are, however,

difficulties associated with the estimation of such breakthrough curves, notably recapturing soil water conditions at discrete depths. The use of breakthrough curves to identify time-variant transit time distributions suggests that an assumed distribution shape may be used to estimate the breakthrough curve within a soil profile.

Here we present a further modification to the StorAge Selection approach to assess temporal variation in soil water mixing as storage dynamics change and derive the ages of soil water using a downward step-wise flux approach (feed-forward) within

upland soils. Secondly, we examine the relationship between the age of water stored in discrete soil depths and the age of water of the whole soil column to identify potential depth-dependent StorAge Selection functions. Thirdly, we identify the source and age of evaporation and root-water uptake water from specific depths with changes in the moisture. We use calibrated soil water isotope simulations with xylem water to identify the time variance of atmospheric waters.


## 2 Theory and Methodology

### 2.1 Feed-forward SAS functions: depth-dependent soil volumes

Conceptualization of catchment-scale flow paths in transit-time studies has typically simplified catchment storage into a single compartment with a single input (rainfall) and two outputs (ET and discharge) (Botter et al., 2011; van der Velde et al., 2012).

Numerous methods have been developed using this conceptualization, including time-variant estimations with StorAge Selection (SAS) functions. The SAS function approach tracks water ages in storage using "age-ranked" storage ($S_T$), which is the cumulative sum of water in storage since the time of rainfall ($T$, absolute age of water) (Botter et al., 2011; van der Velde et al., 2012; Harman, 2015). Values of $S_T$ represent the youngest cumulative sum of water in storage (e.g. $S_T = 100$ mm is the youngest 100 mm in storage). The "age-ranked" storage changes in time with the addition of new precipitation ($J$), and removal

of outflow ($Q$) and evapotranspiration ($ET$) from a storage control volume. Within the SAS framework, water of each age, $T$, is removed from storage using an assumed function ($\omega(S_T)$ or its integral, $\Omega(S_T)$). The function may describe greater movement of young water (e.g. exponential distribution), equal movement of all water ages (random mixing) or greater movement of old water (piston flow).

At scales smaller than catchments, quantification of internal fluxes is more significant for tracer estimated transit times. These

fluxes are dominated by vertical flows ($Q$) determined by the soil structure, and may contain diffusion ($D$) between fast and slow flow domains (Gerke and van Genuchten, 1993; Vogel et al., 2010; Sprenger et al., 2018), evaporation ($E$), and root-uptake ($R$). In many circumstances, unsaturated soil is assumed to have negligible lateral fluxes (e.g. HYDRUS 1-D, Essig et al., 2009). Each of these fluxes ($Q$, $E$, $R$, and $D$) likely change with depth and time due to variations in soil properties and moisture conditions (Fig. 1). The use of SAS functions within soils facilitates the assessment of water ages of various soil

depths and their fluxes. To avoid confusion with the large catchment-scale SAS function approach, the terms of the approach are modified for clarity.





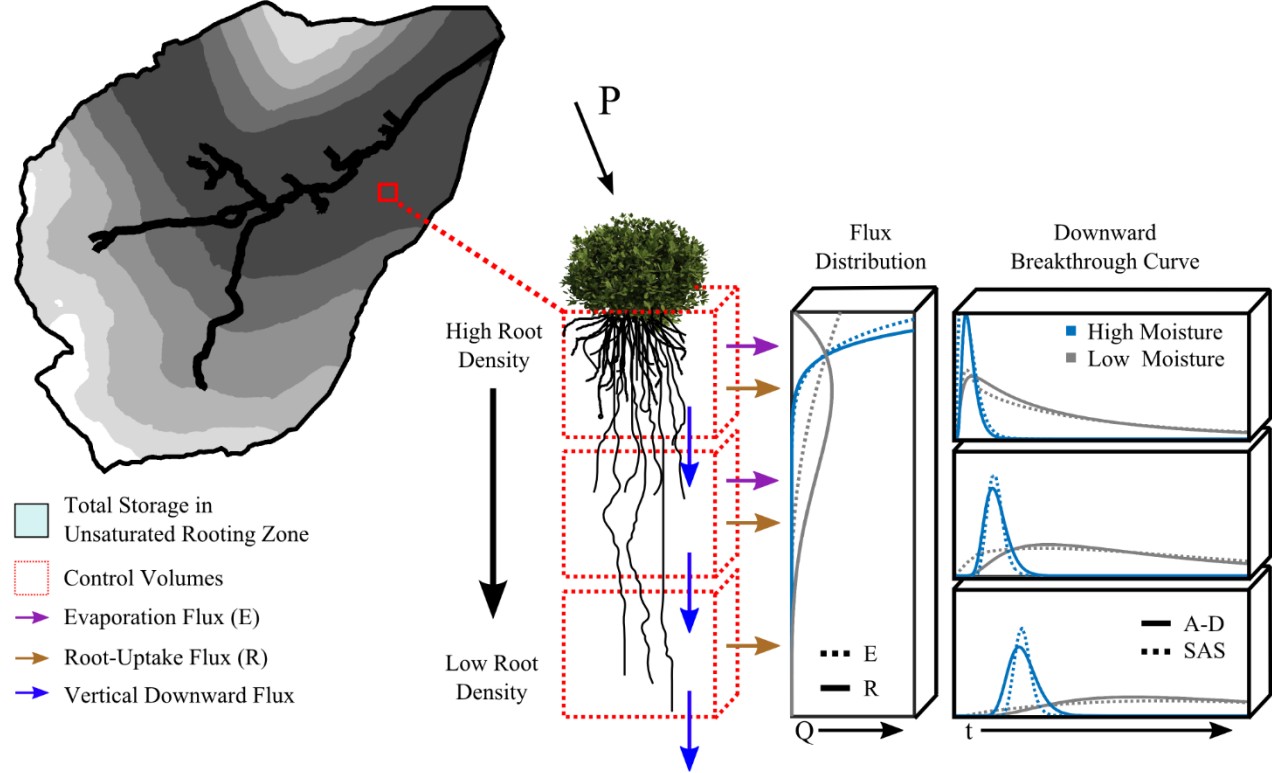

**Figure 1: Conceptual framework of soil water tracer concentration with depth. Root uptake and evaporation fluxes change with depth and soil moisture (coloured arrows). A qualitative comparison of semi-infinite advection-dispersion (A-D) and probabilistic selection (SAS) from storage are shown for each depth**

5    Similar to catchment-scale approaches, defining the storage compartments (termed here control volumes, CV) below the surface is a requirement. Each CV has an equal soil thickness of $\Delta z$. For simplification, upward exchange within the soils is negligible, and the outflow from each CV feeds the CVs below (feed-forward approach, Fig. 1). For each CV, an "age-ranked" storage is defined with respect to the absolute age of water ($S_z(T,t)$), which is a cumulative sum of the time it entered the storage of the CV ($S_{tot}$). Note that $S_{TOT}$ is the water volume within the vertical modelling domain (0 to Z) and $S_{tot}$ is the water

10    volume of a CV ($S_{TOT} = \sum S_{tot}$). In single storage SAS function applications (catchment- or hillslope-scale), the inflow SAS function ($\omega_J(S_z)$) has a uniform age of 0, and the removal of water is derived from the ranked time water entered storage. To remain consistent with this approach, inflow to each CV is assigned a relative age ($\zeta = 0$). In all CVs, the relative age has an inflow distribution related to the absolute age of water ($T$). For example, at the surface CV (depth from 0 - $\Delta z$) the inflow age ($\zeta$) is equal to the absolute age ($T$), while the CV below (depth from $\Delta z$ to $2 \cdot \Delta z$) has a distribution of inflow ages ($\omega_J$) equal

15    to the outflow from the CV above ($\zeta(t) = \omega_J(S_z(T,t,z),t) = \omega_Q(S_z(T,t,z-\Delta z),t)$). For each CV a relative age ranked-storage is defined, $S_z(\zeta,t)$, selecting water by the time it entered the CV. Using this concept, the governing equations using the cumulative SAS function ($\Omega(S_z)$) is defined for the fast and slow flow domains, respectively:



$$\frac{\partial S_z(*,t)}{\partial t} = D(t) \cdot \Omega_D(S_z(*,t),t) - Q(t) \cdot \Omega_Q(S_z(*,t),t) - E_z(t) \cdot \Omega_E(S_z(*,t),t) - R_z(t) \cdot \Omega_R(S_z(*,t),t) - \frac{\partial S_z(*,t)}{\partial *} \quad (1)$$

$$\frac{\partial S_z(*,t)}{\partial t} = -D(t) \cdot \Omega_D(S_z(*,t),t) - R_z(t) \cdot \Omega_R(S_z(*,t),t) - \frac{\partial S_z(*,t)}{\partial *} \quad (2)$$

where $*$ indicates either T (CV from 0 to $\Delta z$) or $\zeta$ (CV below $\Delta z$). For CVs below $\Delta z$, the absolute age of the storage (and fluxes) may be estimated with the relative age ($\zeta$) and its inflow distribution:

$$S_z(T,t) = \int_0^T \int_0^t \left( \omega_J(S_z(T+\zeta,t,z-\Delta z),t) \right) \cdot S_z(\zeta) \cdot d\zeta \cdot dT \quad (3)$$

where the integral from 0 to $t$ cultivates the volume of water with absolute age $T$, while the integral from 0 to $T$ defines the volume of water younger than $T$.

The estimation of water ages in multiple CV also returns a representation of the ages of water in storage ($S_T$):

$$S_T(T,t) = \int_0^Z S_z(T,t,z) \cdot dz \quad (4)$$

This imparts the estimation of the functional form of larger scale SAS functions. The shape of the SAS function is approximated by evaluating:

$$S_z(T,t) = \omega_z(S_T,t) \cdot S_{tot} \quad (5)$$

where $S_z(T,t)$ is estimated from Eq. (3), and $\omega$ is a probability density function with respect to $S_T$ (Eq. 4). This formulation is applicable to fluxes by replacing $S_z(T,t)$ with the age-ranked distribution of the flux, and $S_{tot}$ with the total volume of the

flux from $S_{TOT}$.

**2.2 Conservation of Mass and Isotopic Fractionation**

The impracticality of directly measuring water ages typically results in the use of tracers as tools to approximate water ages (McGuire and McDonnell, 2006). The concentration of tracers is estimated with the same method for water ages; by tracking the concentration of each input of known water age and estimating the selection of concentration in flux using SAS functions

(Eqs. 1 - 5). With stable water isotopes, the concentration refers to the mass of a rare isotope (e.g. $H^2$ or $O^{18}$), which is standardized against the common isotope (e.g. $H^1$ and $O^{16}$). The standardized form is shown as δ. The concentration of a flux ($Q$, $E$, $R$, and $D$) or water storage is estimate by integrating the isotopic composition through all water in storage:

$$\delta_s(z) = \int_0^{S_{tot}} \omega_z(S_z,t) \cdot \delta_z(S_z(T,t),t) \cdot dS_z \quad (6)$$

where $\delta_s$ indicates the isotopic composition of the soil water in a CV, $\omega_z$ is the SAS function used in Eqs. (1 and 2), and $\delta_z$ is

the isotopic composition of water ranked by age ($T$). Special consideration of mass conservation is required for some soil fluxes due to isotopic fractionation effects. Fractionation of stable isotopes during root-uptake is generally assumed to be negligible (Ehleringer and Dawson, 1992), therefore, soil isotopic compositions are unaffected by root-uptake and measured xylem water represents the isotopic composition of soil water. Soil evaporation is more complex since evaporation results in isotopic kinetic fractionation of soil water. The influence of evaporation fractionation for each water age ($T$) is estimated using

mass-balance:



$$S_z(T)\frac{d\delta_z(T)}{dt} + \delta_z(T)\frac{dS_z(T)}{dt} = -Q(T)\cdot\delta_z(T) - D(T)\cdot\delta_z(T) - R(T)\cdot\delta_z(T) - E(T)\cdot\delta_E(T) \qquad (7)$$

where the fluxes ($Q(T)$, $D(T)$, $R(T)$, and $E(T)$) are estimated with the probability functions (Eqs. 1, 2) for specific ages of $T$, and $\delta_E(T)$ is the isotopic composition of evaporative vapour. The Craig-Gordon model (CG, Craig and Gordon, 1965) is the most commonly applied model for the estimation of $\delta_E$. The CG model incorporates aerodynamic resistances with the phase-change fractionation from liquid to vapour ($\alpha^+$), and independently estimates the equilibrium fractionation ($\varepsilon^+$) and kinetic fractionation ($\varepsilon_K$). The liquid-vapour fractionation and equilibrium fractionation are a function of temperature ($T_a$): for deuterium ($\delta^2H$), $\alpha^+ = \exp(1158.8(T_a^3/10^{12}) - 1620.1(T_a^2/10^9) + 794.84(T_a/10^6) - 0.16104 + 2.9992(10^6/T_a^3))$ and for oxygen-18 ($\delta^{18}O$), $\alpha^+ = \exp(-0.007685 + 6.7123(1/T_a) - 1.6664(10^3/T_a^2) + 0.35041(10^6/T_a^3))$ and $\varepsilon^+ = (1-\alpha^+))$ (Horita and Wesolowski, 1994). Kinetic fractionation is a function of aerodynamic diffusion ($n$), atmospheric relative humidity ($h_A$), and the ratio of molecular diffusion coefficients ($C_K$) ($\varepsilon_K = n\cdot C_K\cdot(1-h_A)$). Recent work on soil water evaporative fractionation has modified the CG model to include the effects of soil moisture conditions on the relative humidity and aerodynamic diffusion coefficient ($n$):

$$\delta_E = \frac{1}{(h_z - h_A + \varepsilon_K)}\cdot\left(\frac{\delta_z - \varepsilon^+}{\alpha^+}\cdot h_z - h_A\cdot\delta_A - \varepsilon_K\right) \qquad (8)$$

$$n = 1 - \frac{1}{2}\cdot\left(\frac{\theta(t)-\theta_o}{\theta_{sat}-\theta_o}\right) \qquad (9)$$

where $h_z$ is the relative humidity in the soil, $\delta_A$ is the atmospheric isotopic composition ($\delta_A = (\delta_P - \varepsilon^+)/\alpha^+$) which is a function of the precipitation isotopic composition ($\delta_P$), $\theta_o$ is estimated as the fraction of water that is less driven by gravitational drainage (e.g. field capacity), $\theta(t)$ is the current soil moisture, and $\theta_{sat}$ is saturated soil moisture (Mathieu and Bariac, 1996; Good et al., 2014). In many locations, $h_s$ may be a significant factor by reducing the diffusive flux from the soil to the atmosphere. In wet soils, $h_s$ is at or near 1, and the Eq. (8) is simplified using $h_s = 1$. Finally, the isotopic composition of soil water of age $T$ ($\delta_z(T)$) is identified with the substitution of Eq. (8) into Eq. (7) and solving for $\delta_z(T)$.

Measurement of root-uptake and soil evaporation isotopic compositions integrate the fluxes from multiple depths. In models, this results in the combination of fluxes from multiple CVs. Therefore, the estimation of total isotopic composition of each flux is:

$$\delta_{R\,or\,E}(t) = \int_0^Z p_{R\,or\,E}(z)\cdot\left(\int_0^{S_{tot}}\omega_{R\,or\,E}(S_z,t)\cdot\delta_z(S_z,t)\cdot dS_z\right)\cdot dz\cdot \qquad (10)$$

where $\omega_{R\,or\,E}(S_z,t)$ is SAS function of root-uptake or evaporation for each CV, and $p_{R\,or\,E}(z)$ is the selection of root-uptake or evaporation flux from depth.

## 2.3 Soil water flux estimations

Modelling water and conservative tracer fluxes in soils is challenging due to subsurface heterogeneity, complex mixing, and diffusive exchange between faster and slower flow domains in respective larger and smaller soil pores (Gerke and van Genuchten, 1993; Vogel et al., 2010; Sprenger et al., 2018). The fast flow domain provides more rapid vertical fluxes (downward) while the slow flow domain sustains storage and can exchange water with the fast flow domain in wet soils but





yields no vertical fluxes. We applied a simple storage-discharge relationship to define the volume of the fast and slow flow domains for each CV (Brutsaert and Nieber, 1977). When evaporation and root-uptake from the soils are negligible, the change in soil moisture ($d\theta/dt$) indicates a vertical flux. The storage in each CV is estimated by soil moisture ($S = \theta \cdot \Delta z \cdot \phi$). Since evaporation and transpiration are radiation dependent at our study sites (Wang et al., 2017), an estimation of the storage-

discharge relationship was determined using the recession limb ($-d\theta/dt$) during the night from measured data (see below). Hourly data meeting this criteria were fit with a modified storage-discharge equation:

$$(\theta_i - \theta_o) \cdot \Delta z \cdot \phi = \frac{1}{a} \cdot \frac{1}{2-b} \cdot \left( (\theta_i - \theta_{i+1}) \cdot \Delta z \cdot \phi \right)^{2-b} \qquad (11)$$

where $\bar{\theta}$ is the mean measured soil moisture over the time-step ($\Delta t$), $\phi$ is the porosity, $i$ and $i+1$ indicate the current and next time-step, and $\theta_o$, $a$ and $b$ are fitting parameters. The parameter, $\theta_o$, is the separation of the slow and fast flow domains (see

Eq. 9), and when $\theta < \theta_o$ there is no drainage. Under free-draining conditions $\theta_o$ approaches zero. Eq. (11) is rearranged to solve for the soil moisture (and soil storage) at the next time-step:

$$\theta_{i+1} = \theta_i - \frac{1}{\Delta z} \cdot \left( \left( (\theta_i - \theta_o) \cdot \Delta z \cdot \phi \right) \cdot a \cdot (2-b) \right)^{\frac{1}{2-b}} + \frac{P}{\Delta z \cdot \phi} \qquad (12)$$

During the most dynamic periods of precipitation, Eq. (12) is supplemented with additional downward flow to reduce $\theta_{i+1}$ to the measured $\theta$ ($Q_{add} = (\theta_{i+1} - \theta(t)) \cdot \Delta z \cdot \phi$). The comparison of flux estimation using storage-discharge (Eq. 12) and

HYDRUS 1-D simulations are provided in the Supplementary Material. For simplicity, the diffusion between the fast and slow flow domain was estimated using a linear relationship ($D = V_F \cdot u_D$), where $V_F$ is the volume of the fast flow domain and $u_D$ is the diffusion rate parameter.

With limitations in energy fluxes and soil heat storage measurements, direct estimations of discrete vapour fluxes from depth are difficult. The sources of $E$ and $R$ from depth likely change in time due to water availability (Ogle et al., 2014; Volkmann

et al., 2016; Barbeta and Peñuelas, 2017). To estimate the temporal variability of $E$ and $R$ sources, the beta distribution was assumed to be representative of the probability of water source from each depth within the modelling domain ($0 - Z$):

$$F(z,t) = \int_z^{z+\Delta z} p_F(z,t) \cdot F(t) \cdot dz \qquad (13)$$

$$p_F(z,t) = \frac{\left( \left( \frac{z}{Z} \right)^{k_F - 1} \left( 1 - \frac{z}{Z} \right)^{u_F - 1} \right)}{B(k_F, u_F)} \qquad (14)$$

where $F(z,t)$ is vapour flux (either $E$ or $R$) from a CV, $F(t)$ is the total vapour flux, $p_F$ is the probability of water source

estimated from the surface, and B is the beta function. The total vapour flux of E and R at our studies sites was estimated using the theory of maximum entropy production (Wang et al., 2017). The parameters for the beta distribution can be constant in time (parameter, $k$), or have the potential for time-variance (parameter, $u$) as a function of soil moisture.

## 2.4 Functional forms and parameterisation of SAS functions

The simulation of water ages (Eqs. 1, 2) and isotopic composition (Eqs. 6, 10) is accomplished by updating the age-ranked

storage ($S_z$) and the age-ranked isotopic composition ($\delta_z$) at each time-step with inflow and outflow. The assumed functional



form of the SAS function ($\omega$) for outflow is fundamental to the estimation of water ages. The SAS function may vary from young water- to old water-dominated, which is governed by the soil structure. In this study, the beta distribution is used to identify young- and old-water dominated flow path. Since the beta distribution is defined on the interval [0, 1], $S_z$ should be normalized by the total water in the CV ($P_z = S_z/S_{tot}$ and $P_T = S_T/S_{TOT}$). Similar to other SAS function approaches, the

parameters of the beta distribution vary with changes in storage:

$$\omega_Q(S_z, t) = \frac{(P_z)^{\alpha-1} \cdot (1-P_z)^{\beta(t)-1}}{B(\alpha, \beta(t))} \quad (15)$$

$$\beta(t) = \eta(t) + \tau \quad (16)$$

where B is the beta function, $\alpha$ and $\beta$ are beta distribution parameters, $\eta(t) = \lambda \cdot \left(SM(t) - \min(SM(t))\right)/\sigma_{SM}$, $\lambda$ is a slope parameter for a linear relationship to soil moisture, $\sigma_{SM}$ is the standard deviation of soil moisture, and $\tau$ is the intercept of the

linear relationship to soil moisture. $\beta$ may vary in time with respect to soil moisture conditions (Eq. 16). To avoid a biased assumption that $\beta$ is time-variable, $\lambda$ ranged from [-3 , 3] including zero ($\beta = \tau$). Since $\beta$ must be $\geq 0$ at all times, $\tau$ was greater than zero for all calibrations:

$$\tau = \begin{cases} [0.01, 4] & \lambda \geq 0 \\ abs(\min(\eta(t))) + [0.01, 4] & \lambda < 0 \end{cases} \quad (17)$$

Since $E$, $R$, and $D$ fluxes are calibrated, more simplified forms of the SAS function were applied for (Eq. 10 for $E$ and $R$):

$$\omega_{(E \text{ or } R \text{ or } D)}(S_z, t) = P_z = S_z/S_{tot} \quad (18)$$

which is a uniform distribution (i.e. random selection) and requires no parameterisation.

## 2.5 Model evaluation

Simulations were calibrated by changing parameters for flux equations and SAS functions and were evaluated using an adjusted Nash-Sutcliffe efficiency ($NSE_{adj}$) to best address the isotopic variability in the near surface soil waters. The $NSE_{adj}$ was

modified to incorporate the uncertainty of the soil and xylem duplicate samples and reduce the sensitivity of the simulation on days of higher uncertainty:

$$NSE_{adj} = 1 - \sum\left((1 - p(\delta_s(t), t)) \cdot (\delta_s(t) - \mu_o(t))\right)^2 / \sum(\mu_o(t) - \overline{\mu_o})^2 \quad (19)$$

where $p$ is a normalized kernel density probability (0 to 1), estimated for each given sample day, $\mu_o$ is the mean measured isotopic composition of soil water for a given sample day, and ($\overline{\mu_o}$) is the mean of all measured isotopic compositions (through

all time). The $NSE_{adj}$ was evaluated for simulated daily for $\delta^2 H$, $\delta^{18}O$, and line-conditioned excess (lc-excess, Landwehr and Coplen, 2006). Continuous values of lc-excess $< 0$ indicate kinetic fractionation effects whereas lc-excess $> 0$ indicates differing sources of precipitation. Using the lc-excess reduced bias of the temporally varying precipitation isotopic compositions with the inclusion of long-term isotopic precipitation records:

$$lc - excess = \delta^2 H - a \cdot \delta^{18}O - b \quad (20)$$



where $a$ and $b$ are the linear regression coefficients from the local meteoric water line (LWML). Calibration consisted of 50,000 Monte Carlo simulations, retaining the 100 best simulations. The "best" calibrations were selected using the $NSE_{adj}$ for all measurements ($\delta^2 H$, $\delta^{18}O$, and lc-excess for 5, 10, 15, and 20 cm soil water, and xylem) and a cumulative distribution function (Ala-Aho et al., 2017). Where flux measurements were not available for calibration, a proxy measurement was used

to determine the suitability of the parameters. For example, the parameters for the source of $E$ from soil ($k_E$ and $u_E$, Eqs. 13, 14) were evaluated through the inference of kinetic fractionation (negative lc-excess) within the soil waters. Similarly, the parameters for the source of $R$ ($k_R$ and $u_R$, Eqs. 13, 14) were evaluated by estimating xylem through root-uptake (Eq. 10). Estimated root-uptake profiles had "soft" validation against measured root distribution as the greatest root-uptake will likely occur from the highest densities. For each storage and flux, the simulated water ages of each calibration were compared with

daily median water ages. For the 100 "best" calibrations, a kernel density estimation (KDE) was conducted for each day to reveal the probability of occurrence for the daily median water age.

Due to the small soil water storages, the methodology is sensitive to large influxes of water, particularly when larger than that of soil storage. Under high inflow conditions, the probabilistic selection may result in outflow exceeding the storage volume due to numerical instabilities. To counteract these numerical instabilities, the time-step was varied under high flow conditions

to meet the Courant criteria:

$$C \equiv \frac{Q \cdot \Delta t}{\Delta z} \leq 1 \qquad (21)$$

where $C$ is the Courant number, $Q$ is it the flux, $\Delta t$ is the time-step, and $\Delta z$ is the space-step (Courant et al., 1928).

### 2.6 Study Site

The SAS approach outlined was applied to data collected from two sites representative of the dominant soil-vegetation (podzol-

heather) communities in the Bruntland Burn experimental catchment (3.2 km²) in northern Scotland from October 2015-September 2016. The Bruntland Burn is energy limited, given the latitude (57° 8' N, 3° 20' E), which with high annual relative humidity (>80 %) result in low annual potential evapotranspiration (ET, ~400 mm yr⁻¹) (Tetzlaff et al., 2014). Annual precipitation is ~1000 mm yr⁻¹ with < 5 % snowfall. Previous hydrometeorological studies have partitioned the ET at these podzol-heather units into 56 % transpiration (root-water uptake) and 44 % total evaporation (interception and soil evaporation)

(Wang et al., 2017). The dominant soils in the catchment are peaty podzol, with rankers on the upper hillslopes and peat in the valley bottom (Fig. 2). Soil water sampling and moisture measurements were conducted in two podzol profiles; one overlying sandy-silt drift (Site A), the other over coarse scree deposits (Site B). At each location, heather (Calluna sp. and Erica sp.) shrubs are the dominant vegetation with extensive shallow root systems and, fine roots typically not exceeding 20 cm (Sprenger et al., 2017b). Soil moisture was continuously measured at 10, 20, and 40 cm depths, while water samples were collected once

per month with in each plot at 5, 10, 15, and 20 cm depths. Soil samples (~ 100 g) were taken to the laboratory and water was extracted for isotope analysis by the direct equilibration method (Sprenger et al., 2017a). Heather xylems samples were collected bimonthly during the growing season and water was cryogenically extracted (Geris et al., 2017). Replicate water and





xylems samples (n = 5) were taken for each depth and plant material. Daily rainfall samples were collected for isotope analysis of $\delta^2H$ and $\delta^{18}O$ throughout the study period. All soil and precipitation water samples were analysed for $\delta^2H$ and $\delta^{18}O$ compositions using an off-axis Integrated Cavity Output Spectroscopy (OA-ICOS) (Triple Water-Vapor Isotope Analyzer TWIA-45-EP, Model#: 912-0032-0000 Los Gatos Research, Inc., USA) running in liquid mode with a precision of ± 0.4 ‰

for $\delta^2H$ and ± 0.1 ‰ for $\delta^{18}O$ as given by the manufacturer). Values are expressed in delta per mil (‰) relative to the Vienna Standard Mean Ocean Water standard.

## 3 Results

### 3.1 Study Site

For each profile, the separation of fast and slow flow domains using the storage-discharge equation ($\theta_o$, Eq. 11) suggests a

higher proportion of water stored in the fast flow domain in the upper 10 cm, and a higher proportion of water stored in the slow flow domain at deeper depths (Figs. 2a, 2c). Generally, the greatest uncertainty of the separation of fast and slow flow domains occurs within the upper 5 cm of the soil, which is dominated by organic matter in the O horizon. Soil water samples generally plot along the LMWL throughout the year (Figs. 2b, 2d). Soil waters in the upper 10 cm significantly deviate below the LMWL (negative lc-excess, 95% confidence) during the spring and summer periods indicating evaporative enrichment

(Fig. 2e). Soil waters below 10 cm did not generally deviate from the LMWL (95% confidence) with the exception of a positive deviation (positive lc-excess) above the LMWL in the 20 cm isotopic samples at Site B. The enrichment and variability of $\delta^2H$ and $\delta^{18}O$ was also observed to change with depth, where near surface soils had the highest variability (box plots, Figs. 2b, 2d). Soil water isotopic compositions at both sites depleted during the large winter precipitation events in December 2015 and January 2016, though remained above the LMWL as did the precipitation isotopic compositions (Fig. 3a). Xylem samples

were generally plotted significantly below the LMWL (95% confidence), similar to the upper 10 cm soil samples (Fig. 2e). This is consistent with the higher rooting densities of heather within the upper 10 cm relative to the top 20 cm soil profile as the main source of plant water (Figs. 2a, 2c).





**Figure 2: The location of the two sites in the Bruntland Burn. Dual isotope space for each location is shown for 5 cm (squares), 10 cm (circles), 15 cm (triangles), 20 cm (diamonds), and xylem (stars) waters colour coded in time. The estimated proportions of fast and slow flow domain water for each depth (Eq. 11) and the measured fine root densities of heather (Sprenger et al., 2017b) are also shown.**





### 3.2 Simulated isotopes in soil and xylem waters

The stable isotopes in precipitation at the sites were highly variable, ranging from -170 to 0 ‰ for $\delta^2H$ and from -15 to 15 ‰ for lc-excess (Fig. 3a). Much of the variability occurred during the large precipitation events between December 2015 to January 2016. Simulations of $\delta^2H$ and lc-excess generally captured the measured dynamics for each variable within each of the soil layers of the two profiles, and replicated the greatly damped the isotopic variability relative to the precipitation (Figs 3b - 3i). The mean $NSE_{adj}$ of $\delta^2H$ (lc-excess) in soils (5, 10, 15, and 20 cm) was 0.75 (0.51) and 0.53 (0.34) for Site A and B respectively. The most notable deviation of the simulation from measured composition was the 2015/2016 event for the upper 5 cm soil layer, where the simulations were more enriched than the measured isotopic composition. During the wettest period, there was a slight reduction in $\delta^2H$ uncertainty relative to the dry periods at each depth. The simulated lc-excess followed the general trend of measured lc-excess, with higher lc-excess during the winter months and lower lc-excess during the summer months. However, the simulated lc-excess had much narrower uncertainty bounds relative to range of the lc-excess of daily replicate samples (Figs 3b - 3d, 3f - 3h).





**Figure 3: (a) Rainfall amount and isotopic composition for the study period. (b) 95% confidence bounds for simulated δ2H and lc-excess at 5, 10, 20 cm and xylem waters.**



Soil waters integrated from the upper 20 cm provided a plausible source of root-uptake in the simulation of the xylem $\delta^2H$ (Eq. 10) and lc-excess under varying soil moisture conditions (Figs. 3e, 3i). Both measured and simulated xylem waters ($\delta^2H$) were less variable in composition than the near surface waters (5 cm) despite the high percentage of fine root densities in the upper 5 cm (Figs. 2a, 2c). The simulations of xylem lc-excess were able to capture the bounds of the measured lc-excess, however the daily mean measured lc-excess was much more variable than the simulation (either xylem or soil waters).

### 3.3 Temporal variability of soil water and percolation water ages

The variability of soil moisture and water storage was site dependent, reflecting the different subsurface drainage properties and soil moisture of the soil profiles at each site (Fig. 4a). Differences in drainage properties and soil moisture were reflected in the simulated median water ages for the CVs at each site (Figs. 4b, 4c). At both sites, median water ages in storage were on average < 200 days for all depths in both the fast and slow flow domain, with restricted periods (notably in the spring of 2016) where median water ages in some depths were older than 200 days. For both sites, median water ages in the slow domain were, on average, older than those in the fast domain, and showed less temporal variability. Median water ages in the CV generally increased with depth at Site A, which was most apparent in the slow domain (Fig. 4b). The median soil water ages at the more-freely draining Site B were generally younger than those at their equivalent depths at Site A (Fig. 4c). The younger soil water ages at Site B were also accompanied by much larger variability. For both sites, the mean age of water retained in the top 20 cm were markedly longer than the ages of groundwater recharge fluxes leaving as percolation from the soil profile (bottom of Figs. 4b, 4c). Similar to the $\delta^2H$ simulations, there was lower uncertainty in the mean age estimates during wet conditions, exemplified by higher KDEs (red) and narrower uncertainty bounds (Figs. 4b, 4c).



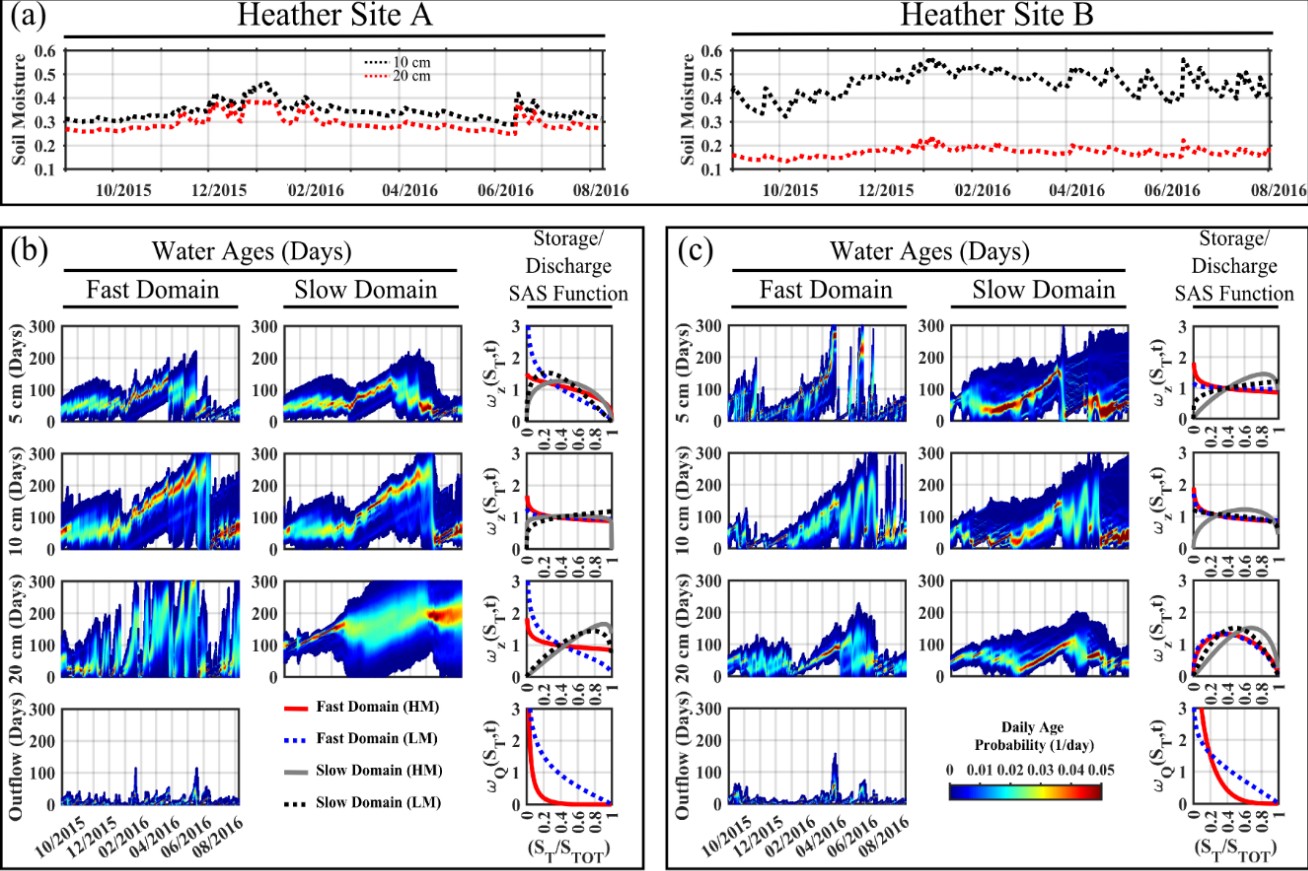

**Figure 4: Median water ages in the fast and slow domain for Site A and Site B at 5, 10, and 20 cm. The water ages of the outflow from the 20 cm bottom boundary are shown at the bottom. The estimated StorAge Selection functions for each flow domain and depth are shown relative to the total storage ($S_T$). HM is the mean SAS function during high soil moisture conditions and LM is the mean SAS function during low soil moisture conditions.**

The SAS function for the water within each CV relative to the total age-ranked storage ($S_T$) was estimated using Eq. (4) and transformed into its probability function ($\omega_z$, Eq. 5). Both sites showed a dominance of the youngest water from $S_T$ in the upper CV fast flow domain storages (Figs. 4b, 4c), through higher dominance of young water in the upper CV occurred at Site A. In the slow flow domain, the dominance of young water was greatly reduced, and exhibited more average water ages. The selection of soil storage water from $S_T$ was variable between sites for the deeper soil CVs, though there was a consistent dominance of younger water within the fast flow domain relative to the slow flow domain. In almost all CVs, there was a slightly higher dominance of younger water within the slow flow domain during dry conditions (black dashed line) relative to wet conditions. Dominant selection of young water as recharge at each site occurred throughout all wetness conditions, and was amplified during the wettest periods (red solid lines and blue dashed lines, bottom of Figs. 4b, 4c). Notably, the differences of dominant water ages between the wettest and driest conditions at Site B were more damped than Site A (Fig. 4c), and had lower dominance of young water.


### 3.4 Evaporation and root-uptake water source and ages

The $E$ and transpiration ($R$) fluxes were quite similar between sites, with slightly higher $E$ at Site A relative to Site B (Figs. 5a, 5b). Calibration of flux source ($k_E$ and $u_E$, Eqs. 10, 11) resulted in estimated high quantities of near surface $E$ flux at both sites (Figs. 5c, 5d). Neither site showed deviation of $E$ source, which was primarily restricted to the CV of the upper 5 cm soil profile. The high proclivity of near surface water for the $E$ source as well as the random selection assumption (Eq. 18), resulted in similar median $E$ ages to the soil ages at 5 cm (Figs. 4b, 4c). During periods of reduced rainfall (spring 2016), the mean age of $E$ water increased for both sites (Figs. 5e, 5f). On average, the median $E$ age was between 50 and 65 days.

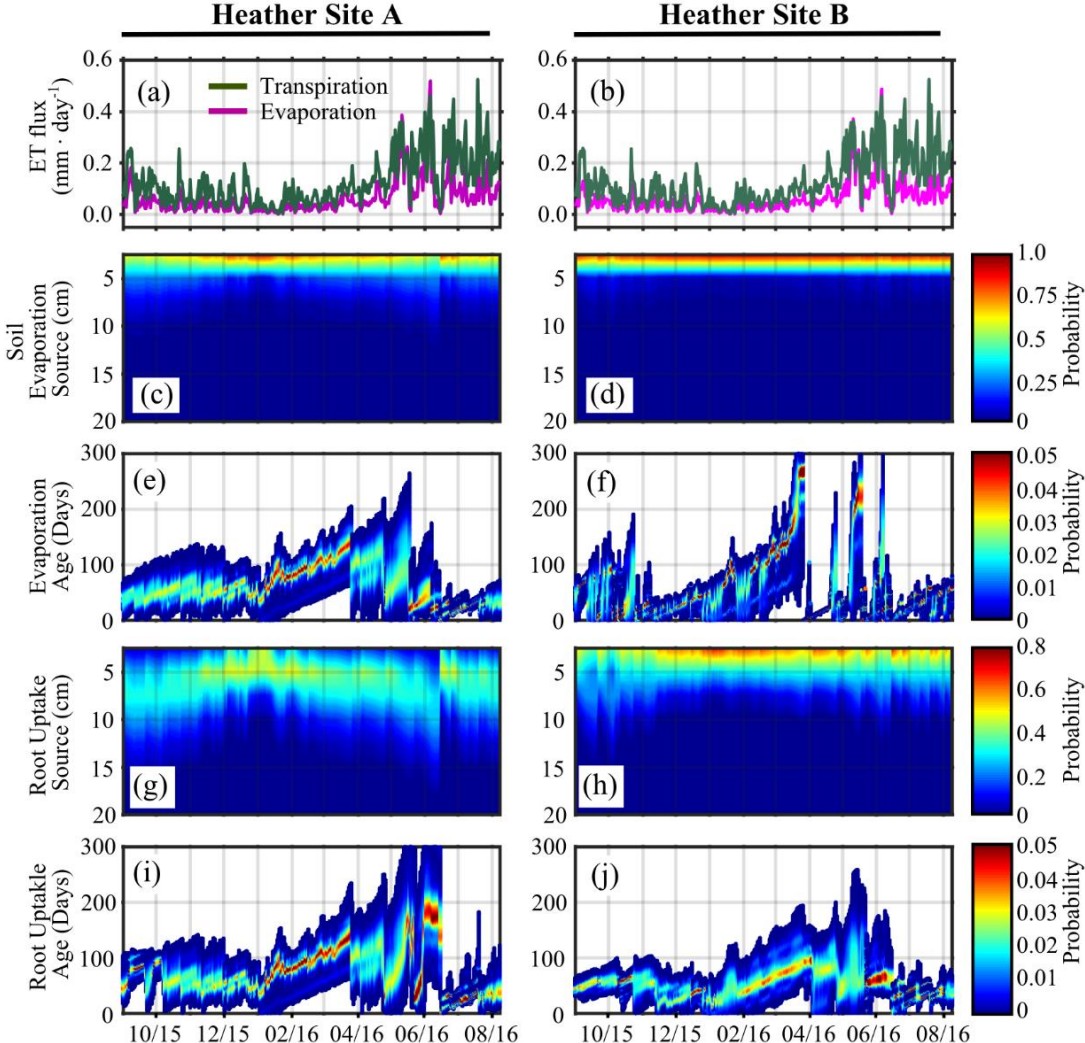

**Figure 5: Simulations and measurements of evaporation and root-uptake at Sites A and B. (a and f) total soil evaporation and transpiration, (b and g) median simulated evaporation water age, (c and h) average probability of evaporation source from depth, (d and i) median simulated root-uptake water age, (e and j) average probability of root uptake source from depth.**





The calibration of the $R$ source ($k_R$ and $u_R$, Eqs. 10, 11) showed temporal variability; during wet periods there was higher selection of $R$ near the surface soil whereas during drier periods the selection of $R$ varied from a wider range of soil depths (Figs. 5g, 5h). In general, the profile of $R$ within each soil profile was relatively similar to the fine root densities measured (Figs. 2a, 2b). Slight deviations between Site A and Site B were noticeable throughout the year, where $R$ at Site B was greater

in the upper 5 cm throughout the year than at Site A. The selection of deeper soil water at Site A relative to Site B resulted in slightly older estimates of median $R$ age ($79.9 \pm 13.8$ and $56.4 \pm 8.9$ days for Site A and B, respectively) relative to median $E$ age. Similar to median $E$ water ages, the median $R$ water age was generally the oldest in the spring of 2016 (Figs. 5i, 5j), which coincided with the period of the least precipitation input (Fig. 3a).

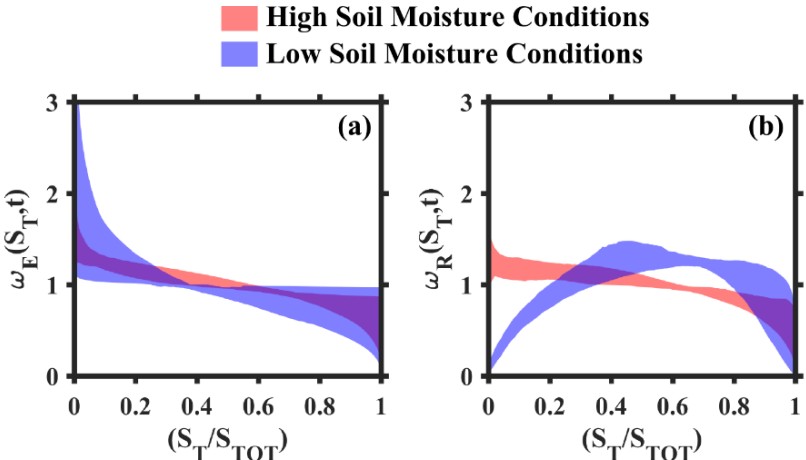

**Figure 6: SAS function bounds for (a) evaporation and (b) root-uptake under high soil moisture (red) and low soil moisture (blue) conditions**

Assessing the depth-dependent $E$ and $R$ profiles with soil storage (Eq. 13) provided a novel opportunity of deriving $E$ and $R$ SAS functions from water ages in storage. Though $E$ and $R$ were simply assumed to be derived from random sampling in each layer, their combination over multiple soil layers provided a more detailed estimate of the median water ages relative to the

total soil water storage column. Regardless of soil saturation, there was a high preference for $E$ to contain the youngest water in the soil column (Fig. 6a). The uncertainty bounds were an accumulation of SAS functions at both sites. The SAS function for $E$ of the wettest conditions (blue bounds, Fig. 6a) had slightly narrower uncertainty bounds than the average driest conditions (red bounds). The SAS function of the driest conditions ranged from very high preference for young water, to near random selection of water in storage. The SAS functions of $R$ were slightly more variable with soil saturation, and showed a

higher affinity for average age water ($S_T/S_{TOT} = 0.5$) under dry conditions (blue bounds, Fig. 6b), and young water under wet conditions (red bounds, Fig. 6b). Unlike the $E$ SAS function bounds, the $R$ bounds were relatively similar for both the wet and dry periods, with the greatest uncertainty surrounding the dry periods and the older water ages.





## 4 Discussion

### 4.1 Ecohydrologic controls of root-uptake on soil water

This study used novel StorAge Selection functions linked with stable water isotopes to identify the source of $E$ and $R$ water while simulating soil water mixing patterns. The approach is an innovative means to address the ecohydrological controls of

atmospheric fluxes on soil water, including the sources of fluxes and the potential for ecohydrologic separation of "energetically available" water for uptake (McDonnell, 2014; Good et al., 2015; Brantley et al., 2017). The simple assumption here that all soil water (both fast and slow flow domain) was available for $R$ was shown, with calibration of $R$ source with depth, to reproduce the xylem isotopic measurements reasonably well. It is notable though, that of the five xylem sample days, one (June 2016, Figs. 3e, 3i) showed isotopic compositions different from either the simulated fast or slow domain isotopic

compositions. This could be indicative of additional undifferentiated water sources in the rooting zone or redistribution of soil water due to root water potential (Domec et al., 2010; Volpe et al., 2013) (e.g. upward flux of water from deeper soils). Alternatively, there is growing evidence that xylem water may be subject to fractionation due to a wide range of biophysical processes that may obscure direct connections with soil water sources (Berry et al., 2017). Some of these biophysical processes may include: effects of evaporative fractionation on xylem isotopic compositions during summer months (Simonin et al.,

2014), the potentially longer residence and transit times of water within the xylem (Brandes et al., 2007), or possible fractionation/discrimination of $O^{18}$ and $H^2$ during root-water uptake (Ellsworth and Williams, 2007; Vargas et al., 2017). While not a large concern in most shrubs, long residence times in larger vegetation (potentially over 20 % of the year, Meinzer et al., 2006) may result in a mixed isotopic composition of root-uptake integrated over numerous days.

Similar to other studies, the $R$ was variable with respect to soil wetness conditions and water availability (Ogle et al., 2014;

Volkmann et al., 2016; Barbeta and Peñuelas, 2017). Drier average conditions at Site A resulted in root-uptake from deeper depths and older water ages, however year-round wet upper soil conditions at Site B resulted in higher $R$ from near surface waters. The development of SAS functions for $R$ to select between young and average water age is a divergence from SAS functions previously used for $ET$ (e.g. Queloz et al., 2015). One essential complexity to define is the depth of the simulation ($Z$). As simulation depth increases, the quantity of older water increases ($P_T(> 0.5)$ increases) since the transit time of

downward flux increases due to further travel distance. The SAS function defined here relates $R$ ages to all water in storage ($S_T$), therefore, as $P_T(> 0.5)$ increases the SAS function ($\omega_R(S_T, t)$) will prefer younger water from $S_T$.

### 4.2 Hydrologic controls of evaporation on soil water

One of the primary difficulties of identifying SAS functions at catchment scales is the shape of the SAS function and the influence of $E$ on the water ages in storage. Calibration revealed highly dominant near surface $E$ (0 - 5 cm soils) which is

consistent with soil $E$ under wet conditions (Sakai et al., 2011; Xiao et al., 2011). The variability of $E$ water age was similar to previous studies in the catchment (Soulsby et al., 2016), with the youngest $E$ water ages during periods of high precipitation. Despite the similarities of the derived SAS function for $E$, the median water age was also similar to previous estimates using





catchment-scale flux tracking on hillslope AET (Soulsby et al., 2016), with a slightly older age at Site A (~15 days older). Notably, the isotopic enrichment of soil water (lc-excess < 0) was simulated at deeper soils depths (Figs. 2b, 2c) though the preference for $E$ sources being primarily constrained to the near surface soil water (upper 5 cm). This reveals a direct influence of $E$ on the upper soils, but an indirect influence of evaporative fractionation on deeper soils. With a higher penchant for near

surface $E$ — where the soils have a much higher proportion of young water — water ages in deeper storages are indirectly affected by the removal of young water via $E$ fluxes. The effect of young water selection by $E$ on increasing water ages in deeper soils has also been observed in lysimeter studies (Queloz et al., 2015). Finally, the overall effect of $E$ and R SAS functions in this study generally show a high tendency for younger water selection by $ET$. However, during dry conditions there is likely a much larger shift towards older waters. During wet conditions this is similar to the shape of $ET$ SAS function

used in previous studies (Harman, 2015; Queloz et al., 2015; Rinaldo et al., 2015; Benettin et al., 2017), while during dry conditions a beta or gamma distribution may be more representative.

### 4.3 Implications of soil water mixing patterns using SAS functions

Soil water modelling using SAS functions showed potential for identifying soil water mixing using stable isotopes of water. The temporally varying beta distribution used in this study was a more dynamic approach to soil water mixing estimates than

a prior mixing assumption (e.g. Timbe et al., 2014) or repeated testing of mixing assumptions at different depths (e.g. Lindström and Rodhe, 1992). The modelling revealed temporal differences in the age of water fluxes, with generally high tendencies for young water to move through the soil profile. These results were similar to lysimeter and hillslope studies, which also showed higher preference of young water movement through the soil under wet conditions relative to drier conditions (Kim et al., 2016; Pangle et al., 2017). The increased preference for young water may be the result of numerous

processes including: the limited additional storage in soils while wet and the freely draining nature of the soil structure (Geris et al., 2015; Sprenger et al., 2017b), or rapid lateral transport due to of rising water tables (Kim et al., 2016; Pangle et al., 2017). The amplified inclination for young water fluxes during wet periods has previously been observed in the stream flow within the catchment (Soulsby et al., 2015). With the selection of young water, the average age of the soil water was older than expected for shallow soils (upper 5 cm), though the median water age through all soil depths was broadly consistent to previous

estimates of the podzols beneath heather elsewhere in the catchment (< 6 months, Tetzlaff et al., 2014). Despite the despite the similar podzolic profiles at the two sites, the soil water movement was somewhat different. The freely-draining deeper soil conditions at Site B was likely the cause of the smaller slow flow domain relative to Site A (Fig. 2), lower soil moisture in deeper soils (Fig. 4a), and younger soil water ages. The higher proportion of water in the slow flow domain further from the surface at Site A is partially due to high water content in the organic surface horizons and a less freely-draining sub-soil than

Site B.

However, some complexities remain with the use of SAS functions within soil columns. For simplification, the slow flow domain modelled here only exchanged with the fast flow domain and did not participate in vertical fluxes. This is due to the uncertainty behind the SAS function characterization and unknown fluxes of the slow domain, but results in simulated water





ages potentially older than with the inclusion of vertical fluxes. Improved characterization of soil fluxes, including the combination of more physically based modelling (e.g. Sprenger et al., 2018) may aid in understanding and properly explaining apparent ecohydrological separation, particularly with the observations in $E$ enrichment. Further aspects that require attention for soil SAS functions are the wettest soil moisture periods. During some of the wettest periods in the simulations, the capability

of the model to accurately simulate the isotopic compositions of near surface water (upper 5 cm) was limited. The depleted isotopic composition measured following the large event was not similar to the isotopic composition of the precipitation of the large event, but rather, the soil isotopic composition may be due to lateral mixing from upslope, which occurs only during rare fully-saturated conditions. It is notable that rainfall totals at this time had estimated return periods > 50 years (Soulsby et al., 2017). Experimental evidence from hillslope studies has shown potential influence of lateral mixing from upslope soils and

changes in infiltration during precipitation event with observed greater influence in the near surface waters (Essig et al., 2009; Morbidelli et al., 2015). Furthermore, a general reduction in the uncertainty of the SAS function was observed for both the age estimation (Fig. 4) and $\delta^2H$ (Fig. 3) during wet conditions, while during dry conditions the uncertainty is the highest. Higher uncertainty during dry conditions is not an anomaly with SAS functions (e.g. Benettin et al., 2017), but exemplifies a general concern for both wet and dry periods regarding the number of data points required to best characterize the SAS function under

extreme conditions.

## 5 Conclusion

The method here provides an adaptation of time-variant StorAge Selection functions for the assessment of soil mixing and a first approximation of evaporation and root-uptake water sources with depth. Time-variant beta distributions identified dominant flow paths of younger water through all soil moisture conditions, though this was highest during wet periods. Time-

variant soil evaporation fluxes showed highest preference for near surface water (0 – 5cm) throughout all soil moisture conditions, consistent with the isotopic fractionation of soil waters. Median evaporation water age was very similar to the median water age near surface soil water (63.3 and 65.3 days, respectively). Root-uptake sources were variable in time, dependent on the soil moisture, and were derived from the near surface (0 – 5 cm) soils under wet conditions, but were derived from deeper soils (5 – 10 cm) as conditions dried. The wider distribution of the root-uptake water resulted in older transpired

water (56 – 79 days) relative to soil evaporation (50 – 65 days). Median percolation water ages were very young (8 to 11 days), with a high preference for young water during wet conditions. The median water age of soil waters is much younger than those estimated on catchment scales due to longer groundwater transit times, however the water ages converge under wet, highly connective catchment conditions. Wider application of the feed-forward SAS function presented here may provide improved understanding the source of both evaporation and root-uptake water while incorporating complex mixing and drainage patterns

of soils in a relatively simple framework.




**Acknowledgements**

We would like to thank the European Research Council (ERC, project GA 335910 VeWa) for funding the project. We would like to thank Matthias Sprenger, Matt Kohn, Hailong Wang, Jonathan Dick and Josie Geris for sample collection and/or isotope analysis. The data used will be available on the University of Aberdeen Public Research Portal (PURE).

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
