# Peer review of "Using StorAge Selection functions to quantify ecohydrological controls on the time-variant age of evapotranspiration, soil water, and recharge"

_Hydrology and Earth System Sciences, 2018_

## Referee Comment (RC1) · Anonymous Referee #1 · 31 Mar 2018

[12pt]article [margin=1in]geometry amsmath,amsthm,amssymb,amsfonts

**Review of "Using StorAge Selection functions to quantify ecohydrological controls on the time-variant age of evapotranspiration, soil water, and recharge"**

Smith et al.

March 31, 2018

**General Comments**

The paper "Using StorAge Selection functions to quantify ecohydrological controls on the time-variant age of evapotranspiration, soil water, and recharge" introduces a novel approach to modeling vertical soil water movement accounting for root-water update, evapotranspiration, and transfer between a slow and fast moving soil domain. The novelty derives from the use of a "feed-forward" StorAge Selection (SAS) function model. Here, the soil column is conceptualized as a stack of control volumes with different SAS function and flux parameters. For each CV, the model simulates fluxes, water ages, and isotopic concentration (accounting for fractionation) in fluxes and storage. The model was calibrated at two field sites with data including observations of xylem water isotopic concentrations and depth profiles of soil moisture and isotopic concentration over a roughly one year period. The simulation results suggest that the water ages are consistent with previous studies, and that there is a generally

strong preference for transport of younger water during wet conditions, which is also consistent with previous studies. Strengths of the paper include the clever and potentially broadly applicable new modeling approach and the informative and artful figures.

I would recommend, however, major revisions to improve confidence in the modeling results that form the basis of the paper's efforts to "investigate water flow in soils and identify soil evaporation and root-water uptake sources from depth" (P1, L12). The revisions should address two separate but related issues.

First, the evaluation of the model is not very compelling and needs to be improved to make the model results more credible. The model seems to have a large number of parameters calibrated to a relatively small number of isotopic measurements with high within-day scatter. The main indicator of model skill shown in the manuscript is an ability to roughly reproduce a seasonal signal in isotopic concentration that dampens with depth (Figure 3). The model also, presumably, simulates soil moisture, but this was not compared to data in the manuscript. The calibration keeps the 100 "best" parameter sets out of 50,000 random samples, which seems to be an arbitrary standard that does not consider the absolute quality of fit between observations and simulations. The final values of the model parameters are not reported, making model performance more difficult to interpret and potentially impossible to reproduce (given the stochastic nature of the calibration). The $NSE_{adj}$ values for the isotopic concentrations are adequate (0.34-0.75), but this is not necessarily compelling given the high number of free parameters. No sensitivity analysis is done to show the importance of different model components in capturing the data. I was left wondering, for example, if a large change in one of the outflow SAS functions (say, in the CV at 10cm) would have an appreciable effect on model performance. If not, then the calibrated values might have a lot of uncertainty that is not presented, and the trends observed in the flux ages might not be

significant. I was also left wondering, for example, if the difference between Site A and Site B SAS functions were significant, or within expected modeling uncertainty bounds.

Some potential ways to improve model evaluation are listed here. (1) The model parameters could be clearly listed with their calibrated values and ranges, to give the reader a sense of the uncertainties. (2) The manuscript could start with a much simpler model and build up to the complex model presented, showing at each step how additional model complexity is justified by the data. (3) The manuscript could report a sensitivity analysis to show how each aspect of the model structure is necessary to describe the data.

Second, the description of the model and underlying theory is at times confusing and seemingly imprecise. For example, the same variable is apparently used for age ranked storage in the slow and fast domain (see equations 1 and 2), some equations seem to be dimensional incorrect (see equation 3), and the CDF and PDF of the SAS function are seemingly confused (see equation 5).

If the authors can make substantial improvements to better describe and evaluate the model, then the results presented in the paper (e.g., the relative ages of different ecohydrological flows, the shape of the different SAS functions and their storage dependence at contrasting sites, the approach to simulating fractionation) could be significant contributions that merit publication.

Many of the the issues described above are listed in more detail with page references in the Specific Comments section.

**Specific Comments**

P1, L16-17: Why do dominant young water fluxes lead to stable soil water ages? I would have thought that would make soil water sensitive to inputs.

P1, L20-21: "More variable" water ages? Meaning 50-65 is more variable than 56-79? The two ranges are not very different.

P2, L14-23: As pointed out, SAS functions have not been used to recover soil water ages at different depths. But there are other "physically-based" models that can could be modified to do that (CATHY, ParFlow, etc). Why focus on SAS functions? A better justification would strengthen the manuscript.

P2, L28-34: It would be helpful to outline the structure of the paper to come: theoretical development followed by case study.

P3, L6-7: The phrase "since the time of rainfall" is a bit vague. Consider rephrasing definition of $S_T$.

P3, L11-13: The parenthetical phrases "exponential distribution", "random mixing", and "piston flow" are apples and oranges. One is a distribution and two are concepts. Consider clarifying.

P4, L14: The text refers to a "distribution of inflow ages $(\omega_j)$...". But the notation $\omega$ is already being used for the pdf form of the SAS function (line 11), and this is a distribution of age-ranked storage, not age, with different units. This is either confusing notation or a conceptual mistake, and should be fixed.
p4, L15: The $\zeta$ is described as being a relative age which presumably has units of time (in p5, L12) but is set equal to the PDF form of the rSAS function $\omega_Q$ in p4,L15, which has units of inverse storage (as shown, for example, in Harman 2015 equation 5). This should be clarified. In general, the proof would be easier to follow if the units (e.g, length, inverse time) were identified when parameters are introduced.

P4, L7-9: The age ranked storage can't be the "cumulative sum of the time", since it has units of storage. It is the volume of storage with age <= T. Also, since this is in terms of "absolute age of water", should it be the time since it entered the vertical modeling domain, and not just the CV?

P4, L16-17 and Equation 1 and 2: The nature of the slow and fast domains was not immediately clear. A few more sentences of explanation would be helpful. Do they represent different conceptual storage volumes with different age ranked storages? Can they be illustrated in Figure 1? Are the left hand sides of equations (1) and (2) really identical? Assuming that they are, then we can set the right hand side of equations (1) and (2) equal, which simplifies to $2 * D * \Omega_D = Q * \Omega_Q + E * \Omega_E$. This suggests that during times when Q and E are zero, then D must be zero. Why so?

P5, Equation (3): It is confusing that $S_z$ is a described here as function of two variables (T,t), one variable ($\zeta$), and three variables (T+$\zeta$, t, and z-$\Delta z$). More notation consistency is needed. In addition, the equation does not seem to be dimensionally correct: the LHS has units of length, and the RHS has units of 1/L times L times T times T, or $T^2$. I think I understand what the authors' are trying to express, but it needs to be more precise.

P5, Equation (5): Again, doesn't seem dimensionally correct. Seems like the CDF (not the PDF) of the rSAS function is needed on the RHS ($\Omega_z$ as defined in line 11)

P6, L18: What is $h_s$ and in which equation is it used? It does not appear in equation 7.

P7, L10: The text states that "under free draining conditions $\theta_0$ approaches zero." Does this mean that $\theta_0$ is time-varying? If so, this should be explained more clearly, since the reader is likely to assume model "parameters" (as it was called in L9) are fixed.

P7, L16: how is $V_F$ calculated? Also, it would be helpful to include an equation for the slow domain volume $V_S$. If there is a unique volume of storage associated with the slow and fast domain, then the volumes have separate age-ranked storages? Equations (1) and (2) suggest they are the same.

P8, L8: What is $SM(t)$?

Equation 15: The paper defines $\omega$ as the PDF SAS function in line 11, P3. Shouldn't the CDF form be used here?

P8, L23: The variable $p$ is the normalized kernel density probability of what?

P9, L10: Please provide a reference for the kernel density estimation technique.

P9, L3-4: One additional sentence on how the "best" calibration was selected would be helpful, with the understanding the reader can refer to the citation for more details.

P9, L7: The phrase ..."by estimating xylem through root-uptake..." is confusing. What does it mean to estimate xylem?

P9, L1-4: The manuscript should describe why this calibration approach it thought to produce meaningful "confidence bounds" (as shown in Figure 3), and explain what the bounds mean. As I understand it, the range of the confidence bounds will approach zero as the number of Monte Carlo simulations goes to infinity (i.e. there will be >100 essentially identical "best" calibrations at a single optimal point in the parameter space), which makes the confidence bounds seem arbitrary and difficult to interpret. In other calibration techniques such as GLUE, the confidence bounds approach finite values as the number of MC simulations gets large, which makes the outcome more easy to interpret. I reviewed the Aho-Aho et al 2017 reference, and did not see this issue addressed.

Section 3.1 and Figure 3: The similarity in the isotopic concentrations observed at site A and B across time and depth (shown in Figure 3) is striking. The values and trends in isotopic concentration seem to be the same at both sites, even if individual values vary a bit. This similarity is unexpected given that Site B is described as more freely draining and has a different soil moisture profile (Figure 4, top panels). It seems important for the manuscript to comment on the similarity and whether there are any significant differences in the data collected from the two sites, since the models are calibrated to this data. Related to this, it also seems important to comment on why the difference in the drainability of the soil at site A and B does not seem to affect the measured isotopic concentrations.

Figure 2: What are the two dotted lines that split the Slow and Fast Domain in figure

(a) and (c)? Also, consider adding political boundaries and labels to the map of the UK, to orientate the reader.

Figure 4: The upper soil moisture plots should be included in the figure description.

P14, L8-9: Here it states the "CVs for each site" are shown in Figure 4. This is the first reference I see to the actual number of CVs modeled. Is the bottom of each CV the number shown in the Y axes of Figure 4? This ambiguity speaks to a wider problem, which is that the number of final model parameters and the calibrated parameter values is not reported. Given the stochastic nature of the calibration, the calibrated parameter values used in the manuscript are needed to reproduce the results.

P17, L5-7: The sentence "The selection of deeper soil water at Site A relative to Site B resulted in slightly resulted in..." raises some of the same concerns described above. First, is the difference between the sites considered significant because the confidence intervals don't overlap? If the calibration used 10,000 monte carlo simulations instead of 50,000, would the confidence intervals be different in a way that could affect the significance test? Also, if the difference really is significant, is the effect of this difference apparent in the measured isotopic data? To convince the reader that these small differences in performance between Site A and B are greater than model uncertainty, it is helpful to show how they arise from the calibration data.

P18, L7-10: "It is notable though, that of the five xylem sample days, one (June 2016, Fig 3i, 3e) showed isotopic compositions different from either the simulated fast or slow domain isotopic concentrations." I was confused by this, because I do not see a uniquely bad fit between the simulation confidence interval and the observations on June 2016 in Figures 3i and 3e. I see that the simulation is not very good that day, but

it is also not very good in 10/15 site B. Also, I don't see any differentiation in this plot between the fast and slow domain. Consider clarifying.

P18, L24-26: The meaning of $P_T(> 0.5)$ is not clear.

P18, L28: The phrase "one of the difficulties of identifying SAS functions at catchment scales is the shape of the SAS function..." seems to be circular logic. Also in general it was relatively difficult to follow the logic of this paragraph. Consider reviewing and clarifying.

P19, L32: Why is it that the median water ages were similar to previous estimates *despite* the similarities of the derived SAS function? I would have thought the median water ages would be similar *because of* similarities in the derived SAS function. A bit confusing.

P19, L2: The "Figs. 2b, 2c" do not show simulated isotopic enrichment. Is this the right figure reference? Also, in general, the sentence starting with "Notably," is confusing. Consider clarifying.

P19, L23: Not clear what is meant by the phrase "with the selection of young water" in the context of the sentence. Consider clarifying.

P20, L11-12: How was "a general reduction in the uncertainty of the SAS function" observed in Figures 4 and 3? As best I could tell, the certainty of the SAS function was not explicitly shown in the figures.

P20, L30: "relatively simple framework"... relative to what? The approach seems fairly complex, even without accounting for lateral fluxes.

**Technical corrections**

P1, L29: "has infer a" is a possible typo.

P9, L22: Possible typo: "results".

P10, L5: Missing an open parenthesis.

P12, L5: possible type / extra word: "the"

P15, L8: possible typo: "through" should be "though"

P19, L21: possible typo

P19, L25: possible typo

Supplemental materials: There seems to be a typo or confusing phrase in the first sentence: "...using Eq. Soil fluxes...".

---

## Referee Comment (RC2) · Anonymous Referee #2 · 1 Apr 2018

In this article, the authors would like to propose a new method to quantify ecohydrological controls based on time-variant water ages. The main novelty of this study is using the mass balance of isotopes from different water sources (rainfall, xylem and soil water) to characterise retention times for different storages. The research questions of the work are interesting. However, (1) the proposed methods are poorly explained, (2) the model assumptions and results are not properly tested and evaluated, and (3) the overall writing can be better. I would like that the authors can evaluate their method by a comprehensive sensitivity study, so that I suggest a major revision.

(1) Poor explanation

[Figure]

The paper is difficult to read because of its inconsistent labelling, poor word choices and bad abbreviations. As a result, the paper cannot explain the method. Here are only few examples to illustrate the problems.

(1a) A label is used for multiple variables. For an example, all the "T"s are confusing.

(1ai) P4 Ln 14 the absolute age (T) (1aii) P5 Ln 25 the isotopic composition of water ranked by age (T) (1aiii) P5 Ln 29 evaporation fractionation for each water age (T)

(1b) Different labels are used for a variable

(1bi) For an example, when Precipitation is defined to be "J" in P3 Ln 9, Figure 1 uses P.

(1c) The word choice can be strange.

(1ci) In P3 Ln15 and Ln18, "Diffusion (D)" is an odd choice. Can it be "infiltration", "recharge" or "surface flow"? Honestly, I don't know. Maybe, it will be clearer if the authors can label it on Figure 1. (1cii) I feel that "control volume" or "CV" is a jargon. Can the author just say a "compartment", "gird", "cell" or "element" instead?

(1d) There are some unnecessary abbreviations

(1di) For an instance, "kernel density estimation" (KDE) only appeared two times in the article. When I read KDE in P14 Ln 18, I needed to try to see what KDE is in P9 Ln10.

(1e) The definitions are bad.

(1ei) I don't understand what "T (CV from 0 to delta) or zeta (CV below delta z)" are in P5 Ln 4, because I don't understand why absolute age (T), relative age (zeta), depth (delta z) and Control volume (CV) are related. (1eii) I don't know how high soil and low soil moistures are defined in Figure 6 All of these make the theory and methodology session difficult to read. The whole article should be revised with a better notation housekeeping.

(2) Technical issues

Overall, the evaluation is weak. In P17 Ln 22, uncertainty is recognised to be important, but nothing has been done in terms of model structures and model parameters. The data quality issues were not explored. At the moment, the results are based on the face values of outputs from a complicated model based on some coarse-temporal data. At a result, the assumptions of the models are poorly validated. In fact, the isotope model results are very bad in Figure 3b-i. How these poor results affect the overall retention results should be quantified by a sensitive study.

(2a) Data issues: The author recognised the data issues in P20 Ln14. However, they did not do much about it. The uncertainty related to data measurements is not quantified.

(2ai) How measurement errors affect the retention results was not studied (2aii) The justification for using lc-excess is weak (P8 Lns 27-28)

(2b) Model issues

At the moment, the authors are subjectively selected its model structure without much evaluation. They need to provide results to show how different variables (e.g. different soil water depths) should be included in the proposed method. The authors should explain how different model structures affect the retention results. Because using isotopes is the main novelty here, I want to see more evaluation for the isotope model selection. I want to see how it was set up. I want to know how different models replacing Equations 8 or 9 can affect the overall retention age results. I also want to see how the sensitivity of the model parameters of Equations 8 or 9 affects the storage retention. Similarly, different distributions should be tests for the hydrology part. For example, in addition to the beta distribution in P7 Ln 24, the author should try gamma and other distributions. In fact, the authors have some ideas about it in P19 Ln 10. The authors should explore the sensitivity of parameters (both constant and time-variance) in Equation 14 (P7 Lns 26-27) The temporal scale of data is very coarse. The model
includes processes across different temporal scales. This kind stiff model has a lot of numerical stability issues because of different scales. The authors recognised some stability problems in P9 Ln 19. Explaining how different processes scales affects the stability of the numerical schemes can be useful information for possible model users to know the technical issues related to the implemented model. The authors should illustrate how their numerical schemes may introduce artefacts. The authors need to try different numerical scheme to illustrate that their parameter estimates are not biased. Overall, the above model issues can be resolved by a comprehensive sensitivity study. Of course, the authors can also do some laboratory or field experiment to validate model assumptions at Heather Site A and B.

(3) Poor writing

The discussion is a bit haphazard. For an instance, I don't understand why the fractionation of xylem water is discussed in P18 Lns 12-18. I am not sure whether the proposed model can address this fractionation issue, or it is a limitation of the proposed model. When the authors discuss ecohydrological and hydrological controls of their sites, they need to link them to their test methods and their results. Currently, Sections 4.1 and 4.2 are like a literature review. The conclusion is very weak. The authors just said "improved understanding" in P20 Ln 29. After the whole study, the authors should be able to articulate their "improved understanding". In short, the authors should be able to frame the overall article better.

---

## Editor Comment (EC1) · N. Romano (Editor) · 4 Apr 2018

I would suggest the authors should post some preliminary responses so as to feed the discussion phase of the journal.

---

## Referee Comment (RC3) · Anonymous Referee #2 · 12 Apr 2018

From the initial response, I am not sure whether the author can revise the manuscript to address the raised issues. Apparently, the authors just want to use the Craig-Gordon model (isotopic model) as a black box, and think that testing the model structure is just secondary to the goals of their study. The authors understand that the selection of distribution is important for their proposed method. They said, 'The use of a beta distribution for evaporation and root-uptake (Equations 13&14) is necessary due to the confined soil depth simulated. The use of a gamma distribution for either flux results in selection of deeper soil water (and isotopic compositions) which were not simulated.' However, it seems that they don't want to show how different distributions for evaporation and root-uptake will affect their results. If the authors do not test and

evaluate their models, I am afraid that the whole time-variant age results can be some unreliable artefacts based on a poor model structure and problematic assumptions. I like this work but a good model evaluation is needed.

———————————————————

---

## Author Comment (AC1) · 12 Apr 2018

The authors thank the reviewers for their comments. As requested by the Editor, we here make some initial responses to the issues raised by. From the review comments that have been received, there seem to revolve around three main concerns: 1) the clarity of the theory section resulting from inconsistent terminology and definitions, 2) the description and execution of the calibration process including the parameterization, and the 3) methods of model evaluation and consideration of uncertainty. Technical Clarity With hindsight, we can see that some additional clarity in the theory section would be helpful. In many of the cases highlighted by the reviewers the (understand-

able) primary confusion is the slight inconsistencies of terminology and definitions through the section. To address this, in the revision, the authors will add a glossary table in an appendix with definitions and units of each term. The inclusion of a glossary will help clarify this issues while simultaneously providing a means to assess the dimensional balance to each equation. Model Calibration The calibration section (Section 2.5 and 2.6) will be revised and updated to provide more detail of the application of the depth-dependent SAS functions to each study location (Site A and Site B). This includes the total number of control volumes (4 depths of 5 cm each, with fast and slow domain) and their parameterisation. This has further led to some confusion about the number of parameters calibrated in the model. For clarity in the revised manuscript, the authors will emphasise that there are 3 parameters for the soil water SAS function (downward flow), and 3 parameters for each the evaporation and root uptake. This effectively leads to a total of 6 SAS function parameters (soil and evaporation SAS functions) to estimate the soil layers. The SAS function parameters are identical for each control volume. $\delta 2H$, $\delta 18O$, and lc-excess is simulated and evaluated in each layer. For a simulation to be considered "behavioural", the parameterization of the soil water SAS function must provide reasonable fits (according to the efficiency criteria) for all control volumes. Hence, the sensitivity of one layer on the next, as suggested by Reviewer 1 is resolved because the sensitivity of each layer to the SAS function parameters is implicit. The authors agree that the inclusion of a sensitivity analysis would help clarify this issue as well as providing additional insights for readers looking to apply this model structure in different regions.

Model Evaluation One of the calibration concerns raised was the number of Monte Carlo parameter sets (50,000) and the number of retained parameter sets (100). The authors agree that the selection criteria was not clearly stated. The following explanation will be included in the revised manuscript to best explain the calibration process. The use of 100 best parameter sets was used due to the initial restrictive minimum criteria threshold (NSE $\geq$ 0.4) for $\delta 2H$, $\delta 18O$, and lc-excess for all control volumes (5, 10, 15, and 20 cm) in addition to the simulated xylem water $\delta 2H$, $\delta 18O$, and lc-excess. The

total number of simulations meeting this criteria at each site was <100. For each site, the number of retained parameter sets was increased to 100 by ranking the minimum efficiency for all control volume simulations (EFF): EFF=minâĄ₂([E_1 ($\delta$ˆ2 H),E_2 ($\delta$ˆ2 H),...E_n ($\delta$ˆ2 H),E_1 ($\delta$ˆ18 O),E_2 ($\delta$ˆ18 O),...E_n ($\delta$ˆ2 H), E_1 (lc-excess),E_2 (lc-excess),...E_n (lc-excess)]) where the subscripts 1 to n are number of control volumes (plus one for xylem water). This ranked efficiency ensures that the remaining retained parameter sets were near the minimum criteria threshold. With the minimum threshold criteria, the uncertainty bounds will not decrease with more Monte Carlo simulation set as suggested by Reviewer 1. Regarding the use of GLUE to evaluate the model performance uncertainty, the author considered this originally but felt that a full GLUE-type analysis of the parameter sets would not provide significantly more information. However, to justify this, we see that further description in the manuscript of the Kernel Probability Density Estimation (KDE) for uncertainty will be necessary to help the reader interpret the model uncertainty. The uncertainty bounds (Figures 3-5) were estimated on daily time-steps by fitting a KDE to the daily simulated isotopic compositions, weighted by the likelihood functions. Figures 4 and 5 show the PDF of the KDE, while Figure 3 shows the resultant 95% bounds of the KDE CDF. Unlike the GLUE approach which uses an empirical CDF (eCDF), the KDE may actually be more beneficial with fewer parameter sets since it interpolates and extrapolates the occurrence of likelihood values (beyond observed likelihood values). In circumstances of models with a smooth likelihood population (e.g. normal, gamma, or exponential distributions), the KDE may converge to the shape of the likelihood population faster than the eCDF (Figure 1 in this comment).

For the figure (Figure 1 in this comment), the mean absolute error is calculated as the difference between the estimated CDF (either the KDE or eCDF) and the known 'population' distribution. Reviewer 2 raised model uncertainty in relation to structural and measurement uncertainty as a weakness. The authors agree that some additional comment on the structural and measurement uncertainty of the model should be added to the manuscript. Firstly, the uncertainty of the isotopic measurements is

implicitly included in the calibration (Equation 19), which uses the statistics of the replicate measurements (mean and standard deviation) of soil and xylem water to inform the goodness-of-fit of the model. Using the standard Nash-Sutcliffe criteria requires the only the mean of the measurements and would likely result in reduced uncertainty bounds. This would occur due to higher sensitivity of the model goodness-of-fit to the mean measurement values. Suggestions to test the model structure is secondary to the goals of this study, and while worth examination in additional studies, cross beyond the scope of this study. Regarding the use of the Craig-Gordon model (isotopic model); this has been widely used in many isotopic modelling studies and has been adapted for isotopic fractionation in soil water. The use of a beta distribution for evaporation and root-uptake (Equations 13&14) is necessary due to the confined soil depth simulated. The use of a gamma distribution for either flux results in selection of deeper soil water (and isotopic compositions) which were not simulated. Hopefully this initial response shows how we can revise the manuscript to address the issues raised by the reviwers.

—————————————————

**Fig. 1.** Comparison of the mean absolute error of the CDF of kernel density estimation and emperical CDF estimation against a known CDF distribution

---

## Author Comment (AC2) · 13 Apr 2018

The authors thank Reviewer 2 for their additional comments. The authors are unsure of what Reviewer 2 means when describing the Craig-Gordon model as a 'blackbox'. While not without its assumptions, the Craig-Gordon model was derived from physically-based processes and is the most widely used isotopic model. Additional isotopic models are generally developed from its framework (eg. see He & Smith, 1999). However, there are numerous methods to estimate the atmospheric parameters of the Craig-Gordon model (e.g. $\alpha$, $\varepsilon$k) which may have an influence on the evaporation fractionation. From these methods (Figure 1 below shows an example of the difference

of two methods used to estimate $\alpha$ in the Craig-Gordon model) it is anticipated that the largest influence on the isotopic fractionation is the selection of the diffusion ratio (Di/D) as well as the turbulence parameter (n). The diffusion ratio has relatively high ranges (0.9757 – 0.9955; Horita et al., 2008), which results in a range in $\varepsilon k$ of 4.5 to 24.9‰ ($\varepsilon k=(1-h)\cdot n \cdot ((D/Di\ )\hat{}n-1)$) assuming n = 1. This corresponds to a maximum range of $\sim$6‰ in $\delta^*$ (directly influences the fractionation in each time-step; see Gibson, 2002) with a relative humidity of 80% (the long-term average at the site) and the conditions shown in Figure 1 (see below). In hindsight to the previous response, the authors should have clearly stated that the parameters of the Craig-Gordon model will also be tested in the proposed sensitivity analysis. As the turbulence parameter (n) was estimated as temporally variable, the sensitivity will be tested with time-invariant parameters (eg. n =0.5 or n=1).

For clarity, the rational for not testing additional distribution shapes is due to the multitude of shapes that have already been tested. From initial testing of the model with other distributions (e.g. uniform, exponential, gamma), it was determined that the use (and selection) of a single distribution was restrictive to the model results. However, additional testing of distributions revealed that the beta distribution, with appropriate parameterisation, could replicate the shapes of other commonly used distributions (see Figure 2 below). As such, the parameterisation of the beta distribution included in the model calibration were; high preference of near surface water ($\beta \gg \alpha$; equivalent to a exponential distribution), uniform selection/random mixing ($\beta = \alpha = 1$; equivalent to a uniform distribution), higher preference of mid-depth waters ($\alpha > 1$, $\beta \gg \alpha$; equivalent to a gamma distribution), and high preference of water from near the bottom of the domain ($\alpha \gg \beta$). Calibration of these parameter ranges ensured that testing of multiple different distribution shapes was conducted. Figure 2 shows the PDF and CDF of an exponential, gamma, and uniform distribution with in addition to the PDF and CDF of beta distributions tested within the model. For each different distribution (i.e. exponential, gamma, and uniform), the beta distribution is shown to imitate the shape quite well. The additional benefit of the beta distribution is the other shapes it may produce (see orange lines, Figure 2). As previously suggested, the authors will conduct a sensitivity analysis of the beta distribution parameters (which changes the shape of evaporation and root-uptake selection). Finally, the parameterization of the beta distribution used for both evaporation and root-uptake fluxes tested of both time-variant and time-invariant conditions ($\lambda$ in Equations 13&14 was permitted to equal 0, time-invariant conditions), therefore this model structure does not make the assumption that uptake is time-variant or time-invariant.

References

Gibson, J. (2002), Short-term evaporation and water budget comparisons in shallow Arctic lakes using non-steady isotope mass balance, Journal of Hydrology, 264(1-4), 242-261, DOI: 10.1016/S0022-1694(02)00091-4

He, H., and Smith, R. B. (1999), An advective-diffusive isotopic evaporation‐-condensation model, Journal of Geophysical Research, 104(D15), 18619–18630, DOI:10.1029/1999JD900335.

Horita, J., Rozanski, K., and Cohen, K. (2008), Isotope effects in the evaporation of water: a status report of the Craig–Gordon model, Isotopes in Environmental and Health Studies, 44(1), 23-49, DOI: 10.1080/10256010801887174

Horita, J., Wesolowski, D.J. (1994), Liquid-vapor fractionation of oxygen and hydrogen isotopes of water from the freezing to the critical-temperature. Geochimica et Cosmochimica Acta, 58, 3425–3437, DOI: 10.1016/0016-7037(94)90096-5

Majoube, M. (1971), Fractionnement en oxygene 18 et en deuterium entre l'eau et sa vapeur. Journal de Chimie Physique, 68, 1423–1436. DOI: 10.1051/jcp/1971681423

[Figure]

[Figure]

**Fig. 1.** a) Estimation of $\alpha$L/V (fractionation parameter) against temperature, b) The estimation of $\delta^*$, c) subplot of plot b

[Figure]

**Fig. 2.** a) The CDF of an exponential, gamma, and uniform distribution with a corresponding beta distribution parameterisation. b) The corresponding PDF of the CDFs in Fig2.a

---

## Referee Comment (RC4) · Anonymous Referee #3 · 9 May 2018

**1   General comments**

This paper examined the water ages of evapotranspiration flux, soil water, and recharge and those time-variability. The potential contributions of this paper are: 1) development of the "feed-forward" model using the SAS function approach, 2) presenting the fascinating isotope dataset, and 3) explaining the differences of age in the fluxes and in the soil and examining those time-variabilities based on the developed model calibrated against the dataset. However, the current manuscript needs significant improvements. First, the benefits of using the SAS function approach in the study are not clearly stated. Second, the model development was not described clearly with the several miss-typed

equations and potential errors. The poor description eventually made it difficult to read the discussion part as the discussion part is mainly written based on the calibrated model. Third, the model evaluation was also not described well and poorly performed. Fourth, several logics in the model (result) interpretation should be better clarified. The four points are described in more detail in the following section.

**2   Specific comments**

**2.1   The use of SAS function approach**

The advantages of using the SAS function approach in this study are not clear (and not stated explicitly). If I understand correctly, the fundamental advantage of the SAS function approach is its capacity of simulating the time-variable transport in a "parsimonious way" (as in Line 16 on Page 2). While the authors may argue that their model is parsimonious (in Line 21 on Page 1), the proposed model has quite a large number of parameters (six). Moreover, there are several assumptions in the form of the SAS functions that are not supported by data. For example, why do the SAS functions for $Q$ (or downward flux) have the same form at each layer? Why are the SAS functions for $D$ uniform? Why are the SAS functions for $ET$ and $R$ at each layer uniform? To this end, I am concerned if the SAS function approach was used because of its arbitrariness on choosing functional forms for the SAS functions (which could be a disadvantage of the approach but allows unpleasant flexibility to a modeler), not its parsimoniousness.

**2.2   Development of the model**

The model (Equations 1–12) is not described well and potentially wrong. My concerns are listed below.

**2.2.1 Equations 1–2**

1. The symbol $S_z$ was used for both fast and slow flow domains. I assumed that the $S_z$ in Equation 1 is the age-ranked storage for the fast domain and that the $S_z$ in Equation 2 is the storage for the slow domain, as it is stated in Lines 16-17 on page 4.

2. No influxes: There are no influx terms in Equations 1 and 2. All storages will be continuously depleted. I believe it is a typo. However, it is important to know if the influxes go either to the fast flow zone or to the slow domain, or if the influxes somehow partitioned into both domains. If the latter is the case, how the partition occurs also needs to be described.

3. Doubled root water uptake: Root water uptake occurs in both domains with the rate of $R_z$. Thus, the total root water uptake from the layer is: $2R_z$.

4. Fraction of root water uptake: What fraction of $R$ was drawn from the slow domain, and what fraction was drawn from the fast domain? How were the fractions determined (or assumed)?

5. Slow domain influxes: As previously stated, the slow flow domain is a source of root water uptake in the model. If there is no influx to the slow domain, the slow domain will be continuously depleted as the net flux through the matrix diffusion $D$ is zero.

6. Evaporation only from the mobile zone: This is also an assumption in the model that needs to be clarified and justified.

7. Not clear age-exchange between the fast and the slow domain: If I understand correctly, there is no net-flux between the fast and slow domains, and there only is age-exchange controlled by the SAS function $\omega_D$, which was assumed as the uniform function in the model. However, the physical meaning of $\omega_D$ is very obscure, and how the model works would be different from the cited papers in Line 16 on page 3.

8. $Q \rightarrow Q_z$

**2.2.2 Equation 3**

It would be surprising if the age-ranked storage can be estimated only using the influx. Don't you need to consider outfluxes and the aging of water inside the storage?

**2.2.3 Equation 4**

Integration is inappropriate. That should be the summation as you did in Line 10 on Page 4.

**2.2.4 Equation 5**

I don't think the $\omega_z$ here is a SAS function but is a cumulative residence time distribution mapped on the age-ranked storage.

**2.2.5 Equation 6**

1. $\delta_z$ should be defined better as it is a function of $S_z$ on the right-hand side term and is a function of $z$ on the left-hand side term.
2. And, again, $\omega_z$ is not a SAS function.

**2.2.6 Equation 7**

The $z$ and $t$ dependencies of the terms are not described well.

**2.2.7  Equations 8–9**

1. It is not described well how the $\delta_E$, estimated from Equation 8, can substitute the one in Equation 7. In Equation 7, $\delta_E$ is a function of $T$, while $\delta_E$ in Equation 8 is not.

2. Potential internal inconsistency: I think that this $\delta_E$ is different from $\delta_E$ estimated using the SAS function. How can you resolve the inconsistency in the model?

3. What are the values of the parameters used? Also, were the same values used for all the layers?

**2.2.8  Equations 11–12**

1. Equation 12 is a water mass balance model using the relationship described in Equation 11. Thus, the introducing sentence which reads "Eq. 11 is rearranged to solve for the ..." is not correct.

2. Potential inconsistency in slow and fast flow domain storage: There are two different slow domain storages in the model: $\theta_0 \Delta z$ (as stated in Lines 9-10 on Page 7) and $S_T(t, t)$ (used in Equation 2). If those are different, this inconsistency should be introduced and treated carefully in the manuscript. Such inconsistency also exists for the fast flow domain.

3. In addition to the above point, I think there are three domains in the model among four available combinations: $\text{fast}_\theta$- $\text{fast}_{S_T}$, $\text{fast}_\theta$- $\text{slow}_{S_T}$, $\text{slow}_\theta$- $\text{fast}_{S_T}$, and $\text{slow}_\theta$-$\text{slow}_{S_T}$, where $\text{fast}_\theta$ is the fast flow domain determined by $\theta$, $\text{fast}_{S_T}$ is the fast flow domain determined by $S_T$, and so on. These four available domains are not described at all in the manuscript, and the manuscript misleads readers by stating that there are only two types of domains.
**2.2.9   Equations 13-14**

The authors stated that the parameter $u$ can be time-variant. However, how $u$ was formulated is not described. There is no $\lambda$ in Equation 13-14, while the authors stated that "$\lambda$ in Equations 13&14 was permitted to equal 0, time-invariant conditions" in Lines 5-6, Page C3 in AC2: Response to Reviewer 2. $\lambda$ was introduced later under Equation 16 but not used in Equation 13-14. Perhaps, an equation similar to Equation 16 is required to formulate $u$ here. Also, it would be arbitrary that how the functional form for $u$ was selected. Can it be justified?

**2.3   Model Evaluation**

**2.3.1   Model performance measure: Equation 19**

1. Was the adjusted NSE newly developed in this study, or are there any references to cite?

2. It is unclear what samples were used in the density estimation. Thus, I had to guess that the replicated samples (n=4) were used to construct the density. It is also not clear what bandwidth was used for the kernel density estimation.

3. Moreover, wouldn't the (perhaps chosen arbitrarily) bandwidth plays an important role in considering the measurement uncertainty in the adjusted NSE? I wonder what the benefit of using the kernel density would be compared to the likelihood functions which considers the measurement uncertainty in a statistically more rigorous way (by using statistics of the measurement such as standard deviation). I don't think the use of the kernel density would be a better way of accounting the uncertainty than using such likelihood functions.

**2.3.2   Parameter estimation**

It is not described well how the 100 parameter sets were chosen using the multiple NSEs (for different types of measurements and different measures). The only description is: "The "best" calibrations were selected using the $NSE_{adj}$ for all measurements (..) and a cumulative distribution function (Ala-Aho et al., 2017)" in Lines 2-4 on Page 9. It is unclear what the "cumulative distribution function" is in this context until one looks at the cited paper. More detailed description is required so that potential readers can grasp what the authors did without looking at the cited paper. (By the way, the description (Equations 6-8) in the cited paper is written with typos, so it was hard to understand the method. Thus, I think the equations should be re-written in this paper). Moreover, selecting the 100 best parameter sets (not 200, 1000, ..) is quite arbitrary, and the arbitrariness makes it difficult to interpret the model's uncertainty estimation.

**2.4   Model interpretation**

This part, mainly the discussion, was very hard to read as I don't have a clear picture of the developed model. Thus, I wrote only a few comments on this part at this stage.

**2.4.1   Time-variability**

The authors stated that "this model structure does not make the assumption that uptake is time-variant or time-invariant" (for example, in Lines 6-7 on Page C3 in AC2: Response to Reviewer 2), and they argued that the model and data supported the time-variant hypothesis as perhaps the NSEs were higher when the time-variability was allowed (when the parameters $u_F$ is time-variant).

I don't agree with the statements for two reasons. First, the criteria for choosing the time-invariant model is unclear. Second, and more importantly, I don't think the authors

have enough data to discuss the time-variability, and the detected time-variability could be an artifact. I will discuss these in more detail in the following.

First, when can you say the beta distribution in Equations 14–15 was time-invariant? When the calibrated parameter (let's say $\lambda$) is "exactly" zero? Or, when the absolute value of $\lambda$ is less than a certain threshold?

Moreover, it seems to me that the isotope dataset (presented in Figure 3) is perhaps not sufficient to test the time-variability. With the above (threshold) issue in mind, perhaps the best way to test the hypothesis on the time-variability is to see how the model works for two different cases: one with the $\lambda$ parameter set to 0 and another by allowing calibration of the parameter. If the authors can identify several periods when the model with $\lambda \neq 0$ captures the observed time-variability, the authors perhaps can say that the time-variable model was required. However, I don't think the dataset is enough to be used for this, and perhaps the model still would do a relatively good job with the $\lambda$ parameter set to 0. Thus, I suggest the authors show the model results with the $\lambda$ parameter set to zero. As the use of NSE would not be sufficient to discuss it, the time-series of model results (similar to Figure 3) should be included (at least in Supplemental material) so that readers can agree to the argument on the time-variability.

**2.4.2 System-scale SAS function for $ET$ and $R$**

It should be described better with an equation of how the SAS functions in Figure 6 were estimated.

**2.4.3 Comparison of the estimated range of water age**

Page 20, Lines 24-25: It seems to me that the ranges overlapped each other quite a lot; thus, it is hard to agree with the statement.

**3 Technical correction**

Page 3, Line 15: The term "diffusion" is too broad here. Please specify.

Page 9, Equation 21: The equation is not about model evaluation. Please consider relocating or removing the equation.

Page 17, Line 14: 'More detailed' is not correct.

Page 17, Line 15: Isn't it median water age in storage not average?

Page 18, Line 23: Possible type: "depth of the simulation"

Page 19, Line 24: "Older than expected": What was the expectation and why?

Page 19, Line 25: Possible typo: "Despite the dispite"

---

## Author Comment (AC3) · 4 Jun 2018

**Reviewer 1 General Comments**

First, the evaluation of the model is not very compelling and needs to be improved to make the model results more credible. The model seems to have a large number of parameters calibrated to a relatively small number of isotopic measurements with high within-day scatter. The main indicator of model skill shown in the manuscript is an ability to roughly reproduce a seasonal signal in isotopic concentration that dampens with depth (Figure 3). The model also, presumably, simulates soil moisture, but this was not compared to data in the manuscript. The calibration keeps the 100 "best" parameter sets out of 50,000 random samples, which seems to be an arbitrary standard that does not consider the absolute quality of fit between observations and simulations. The final values of the model parameters are not reported, making model performance more difficult to interpret and potentially impossible to reproduce (given the stochastic nature of the calibration). The NSEadj values for the isotopic concentrations are adequate (0.34-0.75), but this is not necessarily compelling given the high number of free parameters. No sensitivity analysis is done to show the importance of different model components in capturing the data. I was left wondering, for example, if a large change in one of the outflow SAS functions (say, in the CV at 10cm) would have an appreciable effect on model performance. If not, then the calibrated values might have a lot of uncertainty that is not presented, and the trends observed in the flux ages might not be significant. I was also left wondering, for example, if the difference between Site A and Site B SAS functions were significant, or within expected modeling uncertainty bounds. Some potential ways to improve model evaluation are listed here. (1) The model parameters could be clearly listed with their calibrated values and ranges, to give the reader a sense of the uncertainties. (2) The manuscript could start with a much simpler model and build up to the complex model presented, showing at each step how additional model complexity is justified by the data. (3) The manuscript could report a sensitivity analysis to show how each aspect of the model structure is necessary to describe the data. Second, the description of the model and underlying theory is at times confusing and seemingly imprecise. For example, the same variable is apparently used for age ranked storage in the slow and fast domain (see equations 1 and 2), some equations seem to be dimensional incorrect (see equation 3), and the CDF and PDF of the SAS function are seemingly confused (see equation 5). If the authors can make substantial improvements to better describe and evaluate the model, then the results presented in the paper (e.g., the relative ages of different ecohydrological flows, the shape of the different SAS functions and their storage dependence at contrasting sites, the approach to simulating fractionation) could be significant contributions that merit publication. Many of the issues described above are listed in more detail with

page references in the Specific Comments section.

**Response to Reviewer 1 General Comments**

The authors thank Reviewer 1 for their comments. The revised manuscript will include a model parameter sensitivity analysis to aid with the model evaluation and performance and include the final distributions for calibrated parameters. As discussed in the specific comments below, the selection of the 100 best parameter sets was based on a minimum efficiency criteria rather than an arbitrarily selected number of parameter sets. Some confusion on the number of model parameters will be addressed by listing the calibration parameters as well as the number of minimum efficiency criteria to be met. The model structure was derived to be similar to previous studies, to allow for more direct comparison. To address confusion on the model theory, the methods section will be revised and will include a table of variables for ease of interpretation.

**Reviewer 1 Specific Comments**

**R1C1: P1, L16-17**: Why do dominant young water fluxes lead to stable soil water ages? I would have thought that would make soil water sensitive to inputs.

**Response to R1C1**: The statement was referring to the water ages of the water retained in the soil. The authors will modify this statement to: "*Dominant young water in fluxes through the soil, along with relatively low rainfall intensities, results in shorter retention in the soil of young water and a relatively stable soil retention water age.*"

**R1C2: P1, L20-21**: "More variable" water ages? Meaning 50-65 is more variable than 56-79? The two ranges are not very different.

**Response to R1C2**: The statement will be modified to state that the transpiration is slightly older, rather than variable, than evaporation on average.

**R1C3: P2, L14-23**: As pointed out, SAS functions have not been used to recover soil water ages at different depths. But there are other "physically-based" models that can could be modified to do that (CATHY, ParFlow, etc). Why focus on SAS functions? A

better justification would strengthen the manuscript.

**Response to R1C3**: The authors were not using the SAS functions to recapture infiltration effects, rather, to inform on potential mixing regimes within the soil which affect output fluxes. The use of SAS functions provide a means to simply assess different mixing patterns, and aligns with previous water age methods used within the catchment, and provides consistency in the comparison. Additionally, the assessment of different mixing regimes in more physically based-models requires significantly more parameterization than the use of SAS functions. We will make this clearer in the revision.

**R1C4: P2, L28-34**: It would be helpful to outline the structure of the paper to come: theoretical development followed by case study.

**Response to R1C4**: Thank you for your suggestion. We will modify the manuscript to improve the outline of the manuscript structure. *"We present a further modification to the StorAge Selection approach with the theoretical development and case study of a step-wise approach (feed-forward) with multiple storage volumes."*

**R1C5: P3, L6-7**: The phrase "since the time of rainfall" is a bit vague. Consider rephrasing definition of ST.

**Response to R1C5**: Will be amended to "*the cumulative sum of water in storage, ranked by the elapsed time water has spent in storage*"

**R1C6: P3, L11-13**: The parenthetical phrases "exponential distribution", "random mixing", and "piston flow" are apples and oranges. One is a distribution and two are concepts. Consider clarifying.

**Response to R1C6**: As suggested by the reviewer, the authors will modify the examples to "*The function may describe greater movement of young water (young water preference sampling), equal movement of all water ages (random mixing) or greater movement of old water (piston flow).*"

[Figure]

**R1C7: P4, L14**: The text refers to a "distribution of inflow ages $(\omega_j)$...". But the notation $\omega$ is already being used for the pdf form of the SAS function (line 11), and this is a distribution of age-ranked storage, not age, with different units. This is either confusing notation or a conceptual mistake, and should be fixed.

**Response to R1C7**: While the term used $(\omega_J)$ does have direct relationships with the solution of the SAS function of downward flow from the storage volume above, the reviewer is correct that this notation and definition should be clarified. To better distinguish the distribution of inflow water ages, the authors will change the notation from $\omega_J$ to $w_J$, where $w_J$ represents the backwards transit time of the SAS function of downward flow from the storage above and has units of inverse time.

**R1C8: P4, L15**: The $\zeta$ is described as being a relative age which presumably has units of time (in p5, L12) but is set equal to the PDF form of the rSAS function $\omega_Q$ in p4,L15, which has units of inverse storage (as shown, for example, in Harman 2015 equation 5). This should be clarified. In general, the proof would be easier to follow if the units (e.g, length, inverse time) were identified when parameters are introduced.

**Response to R1C8**: The reviewer is correct, $\zeta$ has units of days, similar to *T*. The equation provided ($\zeta = \omega_J(S_z(T,t,z),t) = \omega_Q(S_z(T,t,z-\Delta z),t))$ should not have included $\zeta$. This will be corrected in the manuscript and the units of $\zeta$ will also be provided. "*relative age ($\zeta = 0$ days)*".

**R1C9: P4, L7-9**: The age ranked storage can't be the "cumulative sum of the time", since it has units of storage. It is the volume of storage with age $<= T$. Also, since this is in terms of "absolute age of water", should it be the time since it entered the vertical modeling domain, and not just the CV?

**Response to R1C9**: That was a mistake, the statement should have read "*cumulative sum of water younger than T*" and will be amended to: "*the cumulative sum of water ranked by the elapsed time it has spent in the modelling domain*"

[Figure]

**R1C10: P4, L16-17** and Equation 1 and 2: The nature of the slow and fast domains was not immediately clear. A few more sentences of explanation would be helpful. Do they represent different conceptual storage volumes with different age ranked storages? Can they be illustrated in Figure 1? Are the left hand sides of equations (1) and (2) really identical? Assuming that they are, then we can set the right hand side of equations (1) and (2) equal, which simplifies to $2 \cdot D \cdot \Omega_D = Q \cdot \Omega_Q + E \cdot \Omega_E$. This suggests that during times when Q and E are zero, then D must be zero. Why so?

**Response to R1C10**: The two equations are not equal, rather, the equations were simplified to try to reduce the number of variables within a more general framework. The authors recognize that this may result in confusion and the equation and text will be modified accordingly. For additional clarification the equations will be presented as:

$$\frac{\partial S_f(\zeta,t,z)}{\partial t} = Q(t, z - \Delta z) + D_{SF}(t, z) \cdot \Omega_D(S_s(\zeta,t,z),t) - E_F(t, z) \cdot \Omega_E(S_f(\zeta,t,z),t) - R_F(t, z) \cdot \Omega_R(S_f(\zeta,t,z),t) - D_{FS}(t, z) \cdot \Omega_D(S_f(\zeta,t,z),t) - Q(t, z) \cdot \Omega_Q(S_f(\zeta,t,z),t) - Q_{FS}(t, z) \cdot \Omega_{Q_{FS}}(S_f(\zeta,t,z),t) - \frac{\partial S_f(\zeta,t,z)}{\partial \zeta}$$

and

$$\frac{\partial S_s(\zeta,t,z)}{\partial t} = Q_{FS}(t, z) \cdot \Omega_{Q_{FS}}(S_f(\zeta,t,z),t) + D_{FS}(t, z) \cdot \Omega_D(S_f(\zeta,t,z),t) - D_{SF}(t, z) \cdot \Omega_D(S_s(\zeta,t,z),t) - E_S(t, z) \cdot \Omega_E(S_s(\zeta,t,z),t) - R_S(t, z) \cdot \Omega_R(S_s(\zeta,t,z),t)) - \frac{\partial S_f(\zeta,t,z)}{\partial \zeta}$$

where the subscripts f and s represent the relative age-ranked storage in the fast and slow domain, respectively, and $D_{SF} = D_{FS}$ for all time-steps and represent the movement of slow domain to fast domain ($D_{SF}$) and fast domain to slow domain ($D_{FS}$). Additionally, the water balance of the soil will be shown:

$$\frac{dV_F(t,z)}{dt} = Q(t, z - \Delta z) - E_F(t, z) - R_F(t, z) - Q_{FS}(t, z) - Q(t, z)$$

$$\frac{dV_S(t,z)}{dt} = Q_{FS}(t, z) - E_S(t, z) - R_S(t, z)$$

where $Q_{FS}$ fills the slow domain ($dV_S/dt = 0$), and $V_F$ and $V_S$ are the fast and slow domain volumes, respectively. For the fluxes in all equations, the subscripts F and S

represent the volume of water from the fast or slow domain, respectively.

**R1C11: P5, Equation (3)**: It is confusing that $S_z$ is a described here as function of two variables ($T, t$), one variable ($\zeta$), and three variables ($T+\zeta, t$ and $z-\Delta z$). More notation consistency is needed. In addition, the equation does not seem to be dimensionally correct: the LHS has units of length, and the RHS has units of 1/L times L times T times T, or T 2 . I think I understand what the authors' are trying to express, but it needs to be more precise.

**Response to R1C11**: The clarification of the definition of the inflow water age distribution will aid with the dimensional confusion. The inflow probability distribution (new term, $w_J$) has units of inverse time ($time^{-1}$), which is a function of absolute water age ($T$), current time-step (t), soil volume (at depth, z), and the time it entered the soil volume (determined via the relative age, $\zeta$). Using the relative age ranked storage of a specific control volume ($S_\zeta(\zeta, t, z)$, units of length, mm), the absolute age ranked storage of a control volume is determined by integrating the inflow probability distribution with the relative age-ranked storage over all relative ages: $S_t(T, t, z) = \int_0^\infty w_J(T, t, z, \zeta) \cdot S_\zeta(\zeta, t, z) \cdot d\zeta$

**R1C12: P5, Equation (5)**: Again, doesn't seem dimensionally correct. Seems like the CDF (not the PDF) of the rSAS function is needed on the RHS ($\Omega_z$ as defined in line 11)

**Response to R1C12**: The authors recognize that the previous terminology was confusing. The equation will be modified to show the CDF. Furthermore, the definition of the term (previously shown as $\omega_z$) will be updated for a more explicit/accurate definition of the distribution. For a given time-step, the cumulative distribution of water in a soil volume is related to the total water in the modelling domain via:$P_{DV}(T, t, z) = ((S_t(T, t, z))/(S_T(T, t))) \cdot V_{tot}(t, z)$, where $V_{tot}$ is the total volume of water in a control volume (a given soil layer).

**R1C13: P6, L18**: What is hs and in which equation is it used? It does not appear in

equation 7. P7, L10: The text states that "under free draining conditions $\theta_0$ approaches zero." Does this mean that $\theta_0$ is time-varying? If so, this should be explained more clearly, since the reader is likely to assume model "parameters" (as it was called in L9) are fixed.

**Response to R1C13**: The text should have used $h_z$ rather than $h_s$ for consistency with the CV terms in the earlier equations. These terms were intended to represent the relative humidity of the soil and will be amended in revision. "*In many locations, $h_z$ may be a significant factor by reducing the diffusive flux from the soil to the atmosphere. In wet soils, $h_z$ is at or near 1, and the Eq. (8) is simplified using $h_z = 1$*". The variable $\theta_o$ is fixed in time and estimated for each control volume. As this was unclear that it was intended for general application it will be removed.

**R1C14: P7, L16**: how is $V_F$ calculated? Also, it would be helpful to include an equation for the slow domain volume $V_S$. If there is a unique volume of storage associated with the slow and fast domain, then the volumes have separate age-ranked storages? Equations (1) and (2) suggest they are the same.

**Response to R1C14**: $V_F$ is estimated using the equations shown in Response to R1C10, which will be clarified in the manuscript. For the conditions present in the study, $V_S$ is constant, defined using $\theta_o$ due to wet soil conditions (i.e. there is always water in the fast domain). The downward flow ($Q(t,z)$) is estimated using the storage discharge relationship: $Q(t,z) = (((\theta(t,z) - \theta_o(z)) \cdot \Delta z \cdot \phi) \cdot a(z) \cdot (2 - b(z)))^{(1/(2-b(z)))}$.

**R1C15: P8, L8**: What is SM(t)?

**Response to R1C15**: SM is defined as soil moisture in the text following the equation (P8 Ln 9-10). For consistency and clarity, this will be changed to $\theta$ (Eqn 11 and 12).: "$\eta(t) = \lambda \cdot (\theta(t) - min(\theta(t)))/\sigma_\theta$), $\lambda$ *is a slope parameter for a linear relationship to soil moisture,* $\sigma_\theta$ *is the standard deviation of soil moisture, and* $\tau$ *is the intercept of the linear relationship to soil moisture.*"

**R1C16**: Equation 15: The paper defines $\omega$ as the PDF SAS function in line 11, P3. Shouldn't the CDF form be used here?

**Response to R1C16**: For simplicity in presenting the distribution, the PDF form ($\omega$) was shown. For consistency with the use of the SAS function in Eqs. 1 and 2, the authors will change the form to CDF:

$$\Omega_Q(S_f(\zeta,t,z),t) = \frac{(B_i((\frac{S_f(\zeta,t,z)}{V_F(t)},\alpha,\beta(t))))}{B(\alpha,\beta(t))}$$

where $B_i$ is the incomplete beta function, $B$ is the beta function, $\alpha$ and $\beta$ are beta distribution parameters, $\eta(t) = \lambda \cdot (\theta(t) - min(\theta(t)))/\sigma_\theta$, $\lambda$ is a slope parameter for a linear relationship to soil moisture, $\sigma_\theta$ is the standard deviation of soil moisture, and $\tau$ is the intercept of the linear relationship to soil moisture."

**R1C17: P8, L23**: The variable p is the normalized kernel density probability of what?

**Response to R1C17**: Apologies, this was misstated in the manuscript. The adjusted NSE used the normal distribution and standard deviation of the samples rather than the kernel density function.

**R1C18: P9, L10**: Please provide a reference for the kernel density estimation technique.

**Response to R1C18**: The kernel density estimation method was previously developed to create probability distributions for atypical shapes (Parzen, 1962). To the authors knowledge, the method of using the kernel density estimation to show daily probabilities is used for the first time in this manuscript. The use of the kernel density approach here was to slight modify the GLUE approach. The estimated kernel density function was weighted by likelihood functions. Due to the smaller number of samples meeting the minimum efficiency criteria, we used the kernel density estimation to approximate the distribution of a larger number of samples (Please see *Initial Response to Reviewers 1 and 2* for an example). The use of kernel density estimation additionally produces

distribution "tails" and theoretically results in more conservative (larger) uncertainty bounds that would not be present when only using the selected parameter sets (i.e. using and empirical CDF).

**R1C19: P9, L3-4**: One additional sentence on how the "best" calibration was selected would be helpful, with the understanding the reader can refer to the citation for more details.

**Response to R1C19**: The parameter sets were based on minimum efficiency criteria of 0.4, rather than arbitrarily selected. Since one location (Site B) had fewer than 100 parameter sets meeting the minimum criteria, the next closest parameter sets to meeting the efficiency criteria were included (minimum efficiency near 0.4). The authors will include a more details explanation of how the parameters were selected, and subsequently ranked using the cumulative distribution function (CDF):

$$n(X) = (\cap_{i=1}^{5} \cap_{j=1}^{3} P_{(i,j)}(X \leq x))/100$$

where $P_{i,j}$ is a CDF for a model layer $i$ with efficiency criteria $j$, and $n(X)$ is the number of simulations meeting the objective ($X \leq x$).

**R1C20: P9, L7**: The phrase ..."by estimating xylem through root-uptake..." is confusing. What does it mean to estimate xylem?

**Response to R1C20**: This was unclear and will be amended to: "...*the simulated root-uptake isotopic composition (Eq. 19) with the parameters for the source of R with depth ($k_R$ and $u_R$, Eqs. 13, 14) were evaluated against measured xylem isotopic composition using the efficiency criteria ($NSE_{adj}$)*"

**R1C21: P9, L1-4**: The manuscript should describe why this calibration approach it thought to produce meaningful "confidence bounds" (as shown in Figure 3), and explain what the bounds mean. As I understand it, the range of the confidence bounds will approach zero as the number of Monte Carlo simulations goes to infinity (i.e. there will be >100 essentially identical "best" calibrations at a single optimal point in the

parameter space), which makes the confidence bounds seem arbitrary and difficult to interpret. In other calibration techniques such as GLUE, the confidence bounds approach finite values as the number of MC simulations gets large, which makes the outcome more easy to interpret. I reviewed the Aho-Aho et al 2017 reference, and did not see this issue addressed.

**Response to R1C21**: As mentioned in Response to R1C19, the 100 parameter sets were chosen based on a set minimum efficiency criteria rather than arbitrarily selected. This will be clarified in the manuscript. Similarly (Response to R1C18), the kernel density approach is used to estimate the confidence bounds in a slightly modified method of the GLUE approach.

**R1C22: Section 3.1 and Figure 3**: The similarity in the isotopic concentrations observed at site A and B across time and depth (shown in Figure 3) is striking. The values and trends in isotopic concentration seem to be the same at both sites, even if individual values vary a bit. This similarity is unexpected given that Site B is described as more freely draining and has a different soil moisture profile (Figure 4, top panels). It seems important for the manuscript to comment on the similarity and whether there are any significant differences in the data collected from the two sites, since the models are calibrated to this data. Related to this, it also seems important to comment on why the difference in the drainability of the soil at site A and B does not seem to affect the measured isotopic concentrations.

**Response to R1C22**: This is a good suggestion which may be addressed in more detail on Figs. 3 and 4. The statistical differences of isotopic compositions may be provided on the figure to show how Site A and B differ with depth. The authors remind the reviewer that the isotopic compositions are bulk soil samples, which include young and old water. While a site may be freely draining, it still retains water which may not mix thoroughly with younger water. This is what the simulations of the sites show. The young water, leaves the soil very rapidly (Fig 4 outflows), which results in very small replenishment of the bulk soil water with young water. The scale on Fig. 4 outflows will

be adjusted to better show the differences of how young water moves through each site.

**R1C23: Figure 2**: What are the two dotted lines that split the Slow and Fast Domain in figure (a) and (c)? Also, consider adding political boundaries and labels to the map of the UK, to orientate the reader

**Response to R1C23**: The dotted lines represent the uncertainty of $\theta_o$ using the iterative solver. These uncertainties are small and do not provide a significant influence on any results. The authors will add a to the figure caption to explain the dotted lines.

**R1C24: Figure 4**: The upper soil moisture plots should be included in the figure description.

**Response to R1C24**: The authors will include the moisture plots in the caption.

**R1C25: P14, L8-9**: Here it states the "CVs for each site" are shown in Figure 4. This is the first reference I see to the actual number of CVs modeled. Is the bottom of each CV the number shown in the Y axes of Figure 4? This ambiguity speaks to a wider problem, which is that the number of final model parameters and the calibrated parameter values is not reported. Given the stochastic nature of the calibration, the calibrated parameter values used in the manuscript are needed to reproduce the results.

**Response to R1C25**: The authors apologize for this confusion as it seems we were not clear in our original description. The soils were discretized into 4 control volumes. Since sampling encompassed the soils within a specific control volume (i.e sampling at 5 cm included samples from 0 to 5cm), the CVs were discretized to include the soils sampled for a given depth. We will include: "*Since the soil samples were an aggregate of water between the soil depths (i.e. soil at 5cm includes soil samples from 0 – 5 cm), the modelled soil layers were discretized into 5cm intervals. From the surface, the layers are named 5 cm (0 – 5 cm), 10 cm (5 – 10 cm), 15 cm (10 – 15cm), and 20 cm (15 – 20 cm).*" The calibrated parameters were previously not included as the

discussion of parameters does not always provide a clear indication of the mechanisms within the system. For this reason, the shapes of the SAS functions were provided as they provide a more meaningful comparison of high soil moisture and low soil moisture conditions at each site. The authors can provide the distribution of best parameter sets for each site in the revision.

**R1C26: P17, L5-7**: The sentence "The selection of deeper soil water at Site A relative to Site B resulted in slightly resulted in..." raises some of the same concerns described above. First, is the difference between the sites considered significant because the confidence intervals don't overlap? If the calibration used 10,000 monte carlo simulations instead of 50,000, would the confidence intervals be different in a way that could affect the significance test? Also, if the difference really is significant, is the effect of this difference apparent in the measured isotopic data? To convince the reader that these small differences in performance between Site A and B are greater than model uncertainty, it is helpful to show how they arise from the calibration data.

**Response to R1C26**: As discussed in Response to R1C19, the best parameter sets were selected based on a minimum efficiency criteria rather than an arbitrarily set number of simulations. This will be clarified in the revised manuscript. As suggested by Reviewer 1 (R2C22), the authors will include some statistical differences between the measured isotopic compositions of the sites in the appendix. The authors will also include statistical differences between the simulated ages of each site in the appendix.

**R1C27: P18, L7-10**: "*It is notable though, that of the five xylem sample days, one (June 2016, Fig 3i, 3e) showed isotopic compositions different from either the simulated fast or slow domain isotopic concentrations.*" I was confused by this, because I do not see a uniquely bad fit between the simulation confidence interval and the observations on June 2016 in Figures 3i and 3e. I see that the simulation is not very good that day, but it is also not very good in 10/15 site B. Also, I don't see any differentiation in this plot between the fast and slow domain. Consider clarifying.

**Response to R1C27**: The large deviation occurs with $\delta^2 H$ (red squares) rather than lc-excess. The difference is much more noticeable for Site B than Site A. In Site A, most of the measured samples are more depleted than the simulation, with the exception of one sample. For the reason that there were no samples for the fast and slow domain independently, the isotopic compositions were only shown as the bulk water (all water in a CV). The authors will modify the statement to indicate that focus should be on $\delta^2 H$ and that the fast and slow domains are not shown on the figure.

**R1C28: P18, L24-26**: The meaning of PT (> 0.5) is not clear.

**Response to R1C28**: The definition of $P_T$ was provided in the methods section (P8 Ln 4), and is effectively the CDF of ST. Therefore, $P_T(0.5)$ is the median age, and $P_T(> 0.5)$ is water older than the median age. The authors recognize that this definition may not have been clear and will modify the description.

**R1C29: P18, L28**: The phrase "one of the difficulties of identifying SAS functions at catchment scales is the shape of the SAS function..." seems to be circular logic. Also in general it was relatively difficult to follow the logic of this paragraph. Consider reviewing and clarifying.

**Response to R1C29**: The authors thank the reviewer for the suggestion and agree that the statement was previously confusing. The statement will be modified to: "*One of the primary difficulties of identifying the temporal variability of flow paths at catchment scales is the shape of the SAS function*" The authors will further revise the discussion section.

**R1C30: P19, L32**: Why is it that the median water ages were similar to previous estimates despite the similarities of the derived SAS function? I would have thought the median water ages would be similar because of similarities in the derived SAS function. A bit confusing.

**Response to R1C30**: This statement will be revised to: "*The use of the temporally*

*variable selection for E yielded little time variance at either site, and resulted in esti-*
*mates of evaporation water age similar to previous catchment-scale flux tracking on*
*hillslope AET. . .*"

**R1C31: P19, L2**: The "Figs. 2b, 2c" do not show simulated isotopic enrichment. Is
this the right figure reference? Also, in general, the sentence starting with "Notably," is
confusing. Consider clarifying.

**Response to R1C31**: This was misstated and should refer to Figs. 3d, 3h and the
simulations at 20cm for $\delta^2 H$ at both sites. The statement will be amended to: "Although
the evaporative fractionation primarily occurred within the upper 5cm, some isotopic
enrichment of deeper soil water was observed in annual cycles of negative lc-excess
(Figs. 3d, 3h)."

**R1C32: P19, L23**: Not clear what is meant by the phrase "with the selection of young
water" in the context of the sentence. Consider clarifying.

**Response to R1C32**: The statement will be modified to: "*The average age of the soil*
*water was older than expected for shallow soils (upper 5cm) due to the preferential*
*selection of young water for downward flux. However, the median water age through*
*all soils depths was broadly consistent to . . .*"

**R1C33: P20, L11-12**: How was "a general reduction in the uncertainty of the SAS
function" observed in Figures 4 and 3? As best I could tell, the certainty of the SAS
function was not explicitly shown in the figures.

**Response to R1C33**: This statement will be modified to: "*A general reduction in the*
*uncertainty of the water age estimation (narrower bands, Fig. 4) and $\delta^2 H$ (narrower*
*bands, Fig. 3) during wet conditions, while during dry conditions the uncertainty is the*
*highest. This may indicate a convergence of the shape of the SAS function during wet*
*conditions while drier conditions are not as sensitive.*"

**R1C34: P20, L30**: "relatively simple framework"... relative to what? The approach

seems fairly complex, even without accounting for lateral fluxes.

**Response to R1C34**: This was intended to state that a non-physically based model provides additional insights into soil water mixing, with shorter calibration run times in a probabilistic framework. We will modify the statement in the revision.

**References**

Harman, C.: Time-variable transit time distributions and transport: Theory and application to storage-dependent transport of chloride in a watershed, Water Resour. Res., 51, 1-30, doi: 10.1002/2014WR015707, 2015.

Parzen, E.: On estimation of a probability density function and mode, Ann. Math. Statist. 33(3), 1065-1076, doi: 10.1214/aoms/1177704472, 1962.

---

## Author Comment (AC4) · 4 Jun 2018

**Reviewer 2 General Comments**

In this article, the authors would like to propose a new method to quantify ecohydrological controls based on time-variant water ages. The main novelty of this study is using the mass balance of isotopes from different water sources (rainfall, xylem and soil water) to characterise retention times for different storages. The research questions of the work are interesting. However, (1) the proposed methods are poorly explained, (2) the model assumptions and results are not properly tested and evaluated, and (3) the overall writing can be better. I would like that the authors can evaluate their method by

Creative Commons CC-BY license logo

a comprehensive sensitivity study, so that I suggest a major revision.

**Response to Reviewer 2 General Comments**

The authors thank Reviewer 2 for their comments. As discussed more in the response to specific comments, the authors will revise the methods section to improve clarity of how the equations were applied and used to solve the water, mass, and age balance equations. Additional justification for the model structure will be incorporated into the methods section to aid in the evaluation. As suggested by the reviewer, a parameter sensitivity analysis will be included to show how well the model performs.

**Reviewer 2 Specific Comments**

**R2C1**: A label is used for multiple variables. For an example, all the "$T$"s are confusing; P4 Ln 14 the absolute age ($T$); P5 Ln 25 the isotopic composition of water ranked by age ($T$); P5 Ln 29 evaporation fractionation for each water age ($T$)

**Response to R2C1**: The original definition provided ($T$ is the absolute water age) is the appropriate definition and the use of $T$ in the other statements was intended to distinguish the general term "age" for absolute vs relative. In our study we define the absolute age as the elapsed time water has spent in the modelling domain, while the relative time is the elapsed time water has spent in a control volume (a given soil layer). As this is unclear, the general term "age" will be accompanied by "absolute" or "relative" where appropriate.

**R2C2**: Different labels are used for a variable. For an example, when Precipitation is defined to be "J" in P3 Ln 9, Figure 1 uses P.

**Response to R2C2**: Figure 1 will be updated for consistency with the label names (eg. P vs J). Since P is later defined as a probability, its use in the figure was a typo.

**R2C3**: The word choice can be strange. In P3 Ln15 and Ln18, "Diffusion (D)" is an odd choice. Can it be "infiltration", "recharge" or "surface flow"? Honestly, I don't know. Maybe, it will be clearer if the authors can label it on Figure 1. I feel that "control

volume" or "CV" is a jargon. Can the author just say a "compartment", "gird", "cell" or "element" instead?

**Response to R2C3**: Figure 1 will also be updated to explicitly show the fluxes within each control volume. This visual aid should help to resolve confusion on the terminology used in the water balance (ie. Diffusion). As suggested by the reviewer, the term "control volume" will be changed to soil layer to be more unambiguous, and more representative of the modelled soil water volume.

**R2C4**: There are some unnecessary abbreviations. For an instance, "kernel density estimation" (KDE) only appeared two times in the article. When I read KDE in P14 Ln 18, I needed to try to see what KDE is in P9 Ln10.

**Response to R2C4**: A table of abbreviations with symbols will be included in the supplementary material. This will aid in the readability of the manuscript given the large number of symbols used for fluxes, water ages, and isotopic compositions. The use of KDE as an abbreviation will be removed.

**R2C5**: The definitions are bad. I don't understand what "T (CV from 0 to delta) or zeta (CV below delta z)" are in P5 Ln 4, because I don't understand why absolute age (T), relative age (zeta), depth (delta z) and Control volume (CV) are related. I don't know how high soil and low soil moistures are defined in Figure 6. All of these make the theory and methodology session difficult to read. The whole article should be revised with a better notation housekeeping.

**Response to R2C5**: The relationship between the absolute and relative age distributions was previously provided in the manuscript (Eq. 3 and P4 Lns 13-16). However, as noted in Response to R1C8, the authors recognize that the units provided in the equation and the definition of $\omega_J$ may lead to confusion. The definition (and relationship between the absolute and relative age) will be modified for clarity.

**R2C6: In P17 Ln 22**: uncertainty is recognised to be important, but nothing has been

done in terms of model structures and model parameters. The data quality issues were not explored. At the moment, the results are based on the face values of outputs from a complicated model based on some coarse-temporal data. At a result, the assumptions of the models are poorly validated. In fact, the isotope model results are very bad in Figure 3b-i. How these poor results affect the overall retention results should be quantified by a sensitive study.

**Response to R2C6**: The authors respectfully disagree that the model results are poor. Also, whilst monthly soil water and xylem is "coarse" it is still an unusually rich data set over a year-long period to test the model with. The calibration of 4 soil layers using one downward flow SAS function (3 parameters) to achieve NSE values > 0.4 for $\delta^2H$, $\delta^{18}O$, and lc-excess shows reasonable functionality of the model to capture feed-forward flow processes. The authors recognize that the abstract nature of probability functions within the model does not aid in interpretation of the parameter sensitivity without a more formal analysis. For this reason, the authors will include a parameter sensitivity analysis in the appendix of the revised manuscript.

**R2C7**: The author recognised the data issues in P20 Ln14. However, they did not do much about it. The uncertainty related to data measurements is not quantified. How measurement errors affect the retention results was not studied. The justification for using lc-excess is weak (P8 Lns 27-28)

**Response to R2C7**: Again, we respectfully disagree with the reviewers comment; the authors directly included the measurement uncertainty in the analysis of the model performance (Eq. 19). The model has high weighting to simulate the temporal periods with low measured variability, and lower weighting when measurement variability is high. In order to directly assess how the measurement error would influence the retention results, recalibration would be required with a different efficiency criteria that does not include the measurement uncertainty. Other evaluation tools to include measurement uncertainty (e.g. Kuppel et al., 2013), use a similar weighting method (e.g. covariance matrices) to reduce the influence of high uncertainty days on the overall model

performance. As this is essentially a sensitivity analysis of how well efficiency criteria perform with input measurements, the authors believe that this analysis would not significantly contribute to the manuscript. Lc-excess was used within this study to ensure that evaporative fractionation was simulated appropriately. Using only $\delta^2H$ and $\delta^{18}O$ as calibration metrics may result in slight bias of each isotope and result in incorrect evaporation estimation. For example, a slight bias for more enriched $\delta^2H$ and slight bias for more depleted $\delta^{18}O$ may provide reasonable efficiency; however, would result in a large difference in the lc-excess from measured values. This additional justification will be added to the manuscript.

**R2C8**: At the moment, the authors are subjectively selected its model structure without much evaluation. They need to provide results to show how different variables (e.g. different soil water depths) should be included in the proposed method. The authors should explain how different model structures affect the retention results.

**Response to R2C8**: As discussed in the initial *Response to Reviewer 2*, the model structure was carefully constructed to mirror data availability. The simulated depths were chosen based on the measured data (at 5cm intervals). Smaller layers would result in more numerical instability and would require smaller time-step to avoid the instabilities (< 1 day). Using larger soil storage would likely result in lost measurement information. For example, at 5cm there is greater evidence of evaporative enrichment than is observed at 10cm. The integration of these two layers would result in the loss of information of greater preference for evaporation to occur from the upper 5cm (Fig 5c and 5d).

**R2C9**: Because using isotopes is the main novelty here, I want to see more evaluation for the isotope model selection. I want to see how it was set up. I want to know how different models replacing Equations 8 or 9 can affect the overall retention age results. I also want to see how the sensitivity of the model parameters of Equations 8 or 9 affects the storage retention.

**Response to R2C9**: The Craig-Gordon model is the standard equation used for isotopic fractionation, which was developed through the use of empirical equations. These equations are all explained through physical principals and changing the use of these equations for this study would require significant experimental work well beyond the scope of the modification of SAS functions presented here. As shown in the initial *Response to Reviewer 2*, the parameters used within the Craig-Gordon model may have some sensitivity within the model results. These parameters will be included in the sensitivity analysis (sensitivity analysis described in Response to R2C6).

**R2C10**: Similarly, different distributions should be tests for the hydrology part. For example, in addition to the beta distribution in P7 Ln 24, the author should try gamma and other distributions. In fact, the authors have some ideas about it in P19 Ln 10. The authors should explore the sensitivity of parameters (both constant and time-variance) in Equation 14 (P7 Lns 26-27). The temporal scale of data is very coarse.

**Response to R2C10**: The selection of the beta distribution as the most representative model structure was based on the flexibility and functionality of the beta distribution to reproduce shapes similar to a linear distribution, uniform distribution, exponential distribution, and gamma distribution while simultaneously producing distribution shapes unique to the beta distribution. The uniqueness of the beta distribution is shown in the initial *Response to Reviewer 2*.

**R2C11**: The model includes processes across different temporal scales. This kind stiff model has a lot of numerical stability issues because of different scales. The authors recognised some stability problems in P9 Ln 19. Explaining how different processes scales affects the stability of the numerical schemes can be useful information for possible model users to know the technical issues related to the implemented model. The authors should illustrate how their numerical schemes may introduce artefacts. The authors need to try different numerical scheme to illustrate that their parameter estimates are not biased. Overall, the above model issues can be resolved by a comprehensive sensitivity study. Of course, the authors can also do some laboratory or field experiment to validate model assumptions at Heather Site A and B.

**Response to R2C11**: The authors are unclear on what the reviewer means by "stiff model" or is intending when suggesting the model includes processes at different temporal scales. The model was run on daily time-steps and uses daily data for the simulations. Numerical instabilities would only occur during the high flow conditions if the time-step is not adjusted. These high flow conditions occurred very rarely, only once throughout the simulation during a 1-in-200 year precipitation event. During these periods, the fluxes were estimated using the daily time-step for consistency with the other time-steps, and discretized to the time-step required to meet the Courant criteria. During model development, sensitivity analysis of the SAS functions with various time-steps was tested to ensure that there would not be a significant deviation to larger time-steps. At sites where the change in soil moisture is greater, significant consideration of the long-term time-step is require to ensure that the large changes in volume over a small time-step are captured. The authors will briefly discuss this consideration in the discussion section (Section 4.3).

**R2C12**: The discussion is a bit haphazard. For an instance, I don't understand why the fractionation of xylem water is discussed in P18 Lns 12-18. I am not sure whether the proposed model can address this fractionation issue, or it is a limitation of the proposed model. When the authors discuss ecohydrological and hydrological controls of their sites, they need to link them to their test methods and their results. Currently, Sections 4.1 and 4.2 are like a literature review. The conclusion is very weak. The authors just said "improved understanding" in P20 Ln 29. After the whole study, the authors should be able to articulate their "improved understanding". In short, the authors should be able to frame the overall article better.

**Response to R2C12**: As shown in Fig 3 for the xylem samples, the simulated xylem samples were not as depleted for lc-excess as the measured compositions (also see P18 Ln. 9-10). Simulations were unable to accurately simulate the fractionation in either the fast or slow domain, and the large fractionation was not observed in the

soil waters. The authors recognize that there are limitations with the current model structure; however, we were demonstrating that there are numerous theories for why the fractionation of xylem water occurs. The occurrence of fractionation in xylem water is not site specific (eg. McCutcheon et al., 2017). The discussion sections (4.1 and 4.2) will be modified to clarify the (1) linkages between the non-physical nature of the model structure to physical mechanisms, (2) further use of SAS functions to test the theories of eco-hydrological separations, and (3) the wider implications of the SAS functions of root-uptake and evaporation (e.g. Fig. 6).

**References**

Kuppel, S., Chevallier, F., and Peylin, P.: Quantifying the model structural error in carbon cycle data assimilation systems, Geosci. Model Dev., 6, 45-55, doi: 10.5194/gmd-6-45-2013, 2013.

McCutcheon, R., McNamara, J., Kohn, M., and Evans, S.: An evaluation of the eco-hydrological separation hypothesis in a semiarid catchment, Hydro. Proc., 31(4), 783-799, doi: 10.1002/hyp.11052

---

## Author Comment (AC5) · 4 Jun 2018

**Reviewer 3 General Comments**

This paper examined the water ages of evapotranspiration flux, soil water, and recharge and those time-variability. The potential contributions of this paper are: 1) development of the "feed-forward" model using the SAS function approach, 2) presenting the fascinating isotope dataset, and 3) explaining the differences of age in the fluxes and in the soil and examining those time-variabilities based on the developed model calibrated against the dataset. However, the current manuscript needs significant improvements. First, the benefits of using the SAS function approach in the study are not clearly stated.

[Figure]

Second, the model development was not described clearly with the several miss-typed equations and potential errors. The poor description eventually made it difficult to read the discussion part as the discussion part is mainly written based on the calibrated model. Third, the model evaluation was also not described well and poorly performed. Fourth, several logics in the model (result) interpretation should be better clarified. The four points are described in more detail in the following section.

**Response to Reviewer 3 General Comments**

The authors thank Reviewer 3 for their comments. The authors will provide further justification for the use of the SAS function approach, and justification of the model structure used. We propose to revise the methods section to improve clarity and ensure that any mis-typed equations are corrected and clarified. Through revision of the methods section, the evaluation of the model performance will be explained and will aid with the interpretation of the model results.

**Reviewer 3 Specific Comments**

**R3C1**: The use of SAS function approach. The advantages of using the SAS function approach in this study are not clear (and not stated explicitly). If I understand correctly, the fundamental advantage of the SAS function approach is its capacity of simulating the time-variable transport in a "parsimonious way" (as in Line 16 on Page 2). While the authors may argue that their model is parsimonious (in Line 21 on Page 1), the proposed model has quite a large number of parameters (six). Moreover, there are several assumptions in the form of the SAS functions that are not supported by data. For example, why do the SAS functions for Q (or downward flux) have the same form at each layer? Why are the SAS functions for D uniform? Why are the SAS functions for ET and R at each layer uniform? To this end, I am concerned if the SAS function approach was used because of its arbitrariness on choosing functional forms for the SAS functions (which could be a disadvantage of the approach but allows unpleasant flexibility to a modeler), not its parsimoniousness.

**Response to R3C1**: The authors agree that the benefit of using SAS functions can be clearer. The benefit of SAS functions provides a method to examine incomplete soil water mixing while also considering the potential non-stationarity of mixing. Within the Bruntland Burn, the non-stationarity of soil water mixing has previously been observed at the catchment scale (Benettin et al., 2017), which suggests that the same non-stationarity may be present within smaller soil volumes. For the number of model outputs that are calibrated, the number of parameters within the model is quite low. For example, time-series are produced for 5cm, 10cm, 15cm, 20cm, and xylem water and evaluated with $\delta^2 H$, $\delta^{18} O$, and lc-excess (15 efficiency criteria). Each efficiency criteria yields different information content. The number of parameters is quite low given the high number of efficiency criteria for calibration (15). In the development of the model, numerous model assumptions were tested. Model structure testing included parameterization of SAS functions at each depth, 3 SAS function parameters at each depth and 12 parameters for 4 soil layers. However, preliminary simulations found little difference in the SAS functions in the layers below 5cm. Additionally, parameterization of SAS functions at each depth provides a high degree of parameter freedom and the model would be significantly over-parameterized. Furthermore, there is little evidence to suggest a significantly different soil types occur within the top 20cm of soil for either site. The SAS function for D was selected as uniform due to insufficient evidence to provide a different distribution. The uniform SAS function required no parameterization. Within model calibration, the mixing (D) was negligible and the soils are not greatly sensitive to the SAS function. D was included to the potential for diffusive mixing in drier soils (Sprenger et al., 2018). As mentioned in the manuscript, evaporation and root-uptake fluxes were assumed to be uniform due to estimation of evaporation and root-uptake source. Assuming a different distribution would over-parameterize the flux and would result in a lack of uniqueness between the SAS function of evaporation (or root-uptake) and the selection of the flux from depth.

**R3C2**: Development of the model. The model (Equations 1–12) is not described well and potentially wrong. My concerns are listed below. The symbol Sz was used for both

fast and slow flow domains. I assumed that the Sz in Equation 1 is the age-ranked storage for the fast domain and that the Sz in Equation 2 is the storage for the slow domain, as it is stated in Lines 16-17 on page 4.

**Response to R3C2**: The authors recognize that the simplification of the equation term names provided may lead to confusion (see Response to R1C10). The manuscript will be modified

$$\frac{\partial S_f(\zeta,t,z)}{\partial t} = Q(t, z - \Delta z) + D_{SF}(t, z) \cdot \Omega_D(S_s(\zeta, t, z), t) - E_F(t, z) \cdot \Omega_E(S_f(\zeta, t, z), t) - R_F(t, z) \cdot \Omega_R(S_f(\zeta, t, z), t) - D_{FS}(t, z) \cdot \Omega_D(S_f(\zeta, t, z), t) - Q(t, z) \cdot \Omega_Q(S_f(\zeta, t, z), t) - Q_{FS}(t, z) \cdot \Omega_{Q_{FS}}(S_f(\zeta, t, z), t) - \frac{\partial S_f(\zeta,t,z)}{\partial \zeta}$$

and

$$\frac{\partial S_s(\zeta,t,z)}{\partial t} = Q_{FS}(t, z) \cdot \Omega_{Q_{FS}}(S_f(\zeta, t, z), t) + D_{FS}(t, z) \cdot \Omega_D(S_f(\zeta, t, z), t) - D_{SF}(t, z) \cdot \Omega_D(S_s(\zeta, t, z), t) - E_S(t, z) \cdot \Omega_E(S_s(\zeta, t, z), t) - R_S(t, z) \cdot \Omega_R(S_s(\zeta, t, z), t)) - \frac{\partial S_f(\zeta,t,z)}{\partial \zeta}$$

where the subscripts f and s represent the relative age-ranked storage in the fast and slow domain, respectively, and $D_{SF} = D_{FS}$ for all time-steps and represent the movement of slow domain to fast domain ($D_{SF}$) and fast domain to slow domain ($D_{FS}$). Additionally, the water balance of the soil will be shown:

$$\frac{dV_F(t,z)}{dt} = Q(t, z - \Delta z) - E_F(t, z) - R_F(t, z) - Q_{FS}(t, z) - Q(t, z)$$

$$\frac{dV_S(t,z)}{dt} = Q_{FS}(t, z) - E_S(t, z) - R_S(t, z)$$

where $Q_{FS}$ fills the slow domain ($dV_S/dt = 0$), and $V_F$ and $V_S$ are the fast and slow domain volumes, respectively. For the fluxes in all equations, the subscripts F and S represent the volume of water from the fast or slow domain, respectively.

**R3C3**: There are no influx terms in Equations 1 and 2. All storages will be continuously depleted. I believe it is a typo. However, it is important to know if the influxes go either to the fast flow zone or to the slow domain, or if the influxes somehow partitioned into both domains. If the latter is the case, how the partition occurs also needs to be

Interactive
comment

described.

**Response to R3C3**: The equation was misstated. See Response to R3C2.

**R3C4**: Doubled root water uptake: Root water uptake occurs in both domains with the rate of Rz. Thus, the total root water uptake from the layer is: 2Rz.

**Response to R3C4**: The total flux, $R$, was weighted by volume between the fast and slow flow domain. The manuscript will be modified to describe this separation. (See Response to R3C2 for the modified equations).

**R3C5**: Fraction of root water uptake: What fraction of R was drawn from the slow domain, and what fraction was drawn from the fast domain? How were the fractions determined (or assumed)?

**Response to R3C5**: See Response to R3C4.

**R3C6**: As previously stated, the slow flow domain is a source of root water uptake in the model. If there is no influx to the slow domain, the slow domain will be continuously depleted as the net flux through the matrix diffusion D is zero.

**Response to R3C6**: See Response to R3C4

**R3C7**: Evaporation only from the mobile zone: This is also an assumption in the model that needs to be clarified and justified.

**Response to R3C7**: The selection of evaporation only from the mobile zone was an assumption that was previously made during other soil water modelling within the basin (Sprenger et al., 2018). For consistency with this approach, the fast domain was assumed to be the only source (most readily available) of water for evaporation. This justification will be incorporated into the manuscript.

**R3C8**: Not clear age-exchange between the fast and the slow domain: If I understand correctly, there is no net-flux between the fast and slow domains, and there only is age-exchange controlled by the SAS function $\omega_D$, which was assumed as the uniform

function in the model. However, the physical meaning of $\omega_D$ is very obscure, and how the model works would be different from the cited papers in Line 16 on page 3.

**Response to R3C8**: The modified Fig. 1 (Response to R2C3) and equations (Response to R3C2) will help to resolve confusion of the fast and slow domain fluxes. While the mechanisms of water exchange in this model are slightly different from the cited papers, the concepts of exchange are similar. The meaning behind the diffusion is the concentration gradient of isotopes between the fast and slow domain. In steady-state conditions with no external fluxes, the isotopic composition in the fast and slow domains approach the same value. In all cases, the diffusion between the fast and slow domain was minimal and exchange was almost entirely limited to the dispersion term.

**R3C9: Equation 3**: $Q \rightarrow Q_z$ It would be surprising if the age-ranked storage can be estimated only using the influx. Don't you need to consider outfluxes and the aging of water inside the storage?

**Response to R3C9**: Equation 3 only describes the translation of the relative water ages of each CV into the absolute ages. The solution of equations 1 and 2, which utilize the inflow and outflow water ages, provide the relative water ages for each CV. This will be clarified in the revision.

**R3C10**: Equation 4 Integration is inappropriate. That should be the summation as you did in Line 10 on Page 4.

**Response to R3C10**: As the reviewer has suggested, for the volumes considered the summation of each CV is simpler to describe and will be modified within the manuscript.

**R3C11: Equation 5**: I don't think the $\omega_z$ here is a SAS function but is a cumulative residence time distribution mapped on the age-ranked storage.

**Response to R3C11**: The authors thank the reviewer for their suggestion. The use of the term "SAS function" was loosely used to describe how water from each soil layer

may be "selected" from the total storage volume. The revision will term the distribution as the "cumulative residence time of layer relative to the soil water age-ranked storage"

**R3C12**: Equation 6 1. $\delta_z$ should be defined better as it is a function of $S_z$ on the right-hand side term and is a function of z on the left-hand side term. 2. And, again, $\omega_z$ is not a SAS function.

**Response to R3C12**: $\omega_z$ was intended to represent the SAS function of a particular flux rather than the definition provided in Eq 5. The manuscript will be amended to:

$$\delta_{*,ForS}(t,z) = \int_0^{V_{ForS}} (\omega_*(S_{ForS}(T,t,z),t,z) \cdot \delta_{ForS}(S_{ForS}(T,t,z),t,z) \cdot dS_{ForS}$$

where * indicates the type of flux leaving storage ($Q$, $D$, $E$, or $R$), $S_F$ and $S_S$ indicate the absolute age-ranked storage in the fast and slow flow domains, respectively, $\delta_F$ and $\delta_S$ are the absolute age-ranked isotopic composition of the fast and slow domain, respectively, and $V_F$ and $V_S$ indicate the total volume of the fast and slow flow domains, respectively. The units of $\omega$ are inverse storage (1/mm), which is integrated with respect to storage.

**R3C13**: Equation 7 The z and t dependencies of the terms are not described well.

**Response to R3C13**: This equation will be modified and defined with respect to $T$, t, and z for each flux, for example for the fast flow domain:

$$\frac{dS_F}{dT}\frac{d(d\delta_F(S_F,t,z))}{dtdT} + \frac{d\delta_F(S_F,t,z)}{dT}\frac{d(dS_F)}{dtdT} = q(T,t,z-\Delta z) \cdot \frac{d(\delta_F(S_F,t,z-\Delta z))}{dT} + d_{SF}(T,t,z)\cdot$$

$$\frac{d(\delta_S(S_S,t,z))}{dT} - q(T,t,z) \cdot \frac{d(\delta_F(S_F,t,z))}{dT} - d_{FS}(T,t,z) \cdot \frac{d(\delta_F(S_F,t,z))}{dT} - r_F(T,t,z) \cdot \frac{d(\delta_F(S_F,t,z))}{dT}$$

$$-e_F(T,t,z) \cdot \frac{d(\delta_e(S_F,t,z))}{dT}$$

where $\delta_F(S_F,t,z)$ and $\delta_S(S_S,t,z)$ are the absolute age-ranked isotopic composition of the fast and slow domains, respectively at soil layer z, $S_F$ and $S_S$ indicate the absolute age-ranked storage in the fast and slow flow domains, respectively, $q(T,t,z-\Delta z)$ is the downward flux from the storage above with absolute age $T$, and $dS_F$, $dF_S$, $r_F$,

and $e_F$ are the fluxes in/out of storage (diffusion, root-uptake, and evaporation, respectively) with absolute age $= T$. The parenthesis of the absolute age-ranked storage (e.g. $S_F(T, t, z) = S_F$) have been removed for clarity of the equation. The isotopic compositions were evaluated for each value of $T$ (i.e. $S_F/dT$).

**R3C14**: Equations 8–9. It is not described well how the $\delta_E$, estimated from Equation 8, can substitute the one in Equation 7. In Equation 7, $\delta_E$ is a function of $T$, while $\delta_E$ in Equation 8 is not.

**Response to R3C14**: The authors will add additional detail on the application of $\delta_E$ within equation 7. Equation 8 is a function of $T$ as it has $\delta_z$. For further clarification, Equation 8 will be modified to (shown here for the fast flow domain):

$$\frac{d(\delta_e(S_F, t, z))}{dT} = \frac{1}{h_z(t) - h_A(t) + \varepsilon_K(t, z)} \cdot \left( \frac{\frac{d(\delta_F(S_F, t, z))}{dT} - \varepsilon^+(t)}{\alpha^+(t)} \right) \cdot (h_z(t) - h_A(t) \cdot \delta_A(t) - \varepsilon_K(t, z))$$

This indicates that $h_z$, $h_a$, and $\delta_A$ are functions of time, $\varepsilon_K$ is a function of time and changes with the soil depth, and the estimation of $\delta_e$ is dependent on the soil isotopic composition from Eq. 7. The parenthesis of the absolute age-ranked storage (e.g. $S_F(T, t, z) = S_F$) have been removed for clarity of the equation.

**R3C15**: Potential internal inconsistency: I think that this $\delta_E$ is different from $\delta_E$ estimated using the SAS function. How can you resolve the inconsistency in the model?

**Response to R3C15**: The equation (Eq 10) will be modified in the manuscript to reduce confusion of the terms. Since $\delta_E$ is not measured the equation will be updated to reflect root-uptake:

$$\delta_R(t) = \sum_{z=i \cdot \Delta z} (\int_{z-\Delta z}^z f_R(z) dz) \cdot [\frac{V_F}{V_{tot}} \cdot (\int_0^{V_F} \omega_R(S_F, t) \cdot \delta_F(S_F, t, z) \cdot dS_F) + \frac{V_S}{V_{tot}} \cdot (\int_0^{V_S} \omega_R(S_S, t) \cdot \delta_S(S_S, t, z) \cdot dS_S)]$$

where $\delta_R$ indicates the total isotopic composition of a vapour flux ($E$ or $R$), where the summation increases with steps of $\Delta z$ evaluated at $i \in 1 \cdots N$ (N number of soil layers) until the maximum modelling domain depth of Z, $f_R$ is the probability distribution of

root-uptake from depth with units of 1/depth evaluated for a specific soil layer, and the combination of isotopic composition of fast and slow flow domain are volume weighted.

**R3C16**: What are the values of the parameters used? Also, were the same values used for all the layers?

**Response to R3C16**: The only parameter within equations 8 and 9 is the $C_K$, which is the ratio of diffusivities of H2/H1 and O18/O16. The remaining values in the equations are measured (i.e. humidity, temperature, soil moisture) or estimated ($\delta_A$, and $\theta_o$). Since n is dependent on $\theta_o$ and $\theta$ (which were different for each soil layer), the value of n changed between layers. Although the differences of n between layers were quite small. $\theta_o$ was estimated using equations 11 and 12. The revised manuscript will rearrange the equation to define $\theta_o$ prior to Eqs 8 and 9.

**R3C17**: Equations 11–12. Equation 12 is a water mass balance model using the relationship described in Equation 11. Thus, the introducing sentence which reads "Eq. 11 is rearranged to solve for the . . ." is not correct.

**Response to R3C17**: The statement will be amended to better describe the evaluation for equation 12.

**R3C18**: Potential inconsistency in slow and fast flow domain storage: There are two different slow domain storages in the model: $\theta_o\Delta z$ (as stated in Lines 9-10 on Page 7) and $S_T(T, t)$ (used in Equation 2). If those are different, this inconsistency should be introduced and treated carefully in the manuscript. Such inconsistency also exists for the fast flow domain.

**Response to R3C18**: The revised manuscript will modify this terminology (see Response to R3C2 for the terms that will be used).

**R3C19**: In addition to the above point, I think there are three domains in the model among four available combinations: fast$\theta$- fastST , fast$\theta$- slowST , slow$\theta$- fastST , and slow$\theta$- slowST , where fast$\theta$ is the fast flow domain determined by $\theta$, fastST is the

fast flow domain determined by $S_T$ , and so on. These four available domains are not described at all in the manuscript, and the manuscript misleads readers by stating that there are only two types of domains. The authors stated that the parameter u can be time-variant. However, how u was formulated is not described. There is no $\lambda$ in Equation 13-14, while the authors stated that "$\lambda$ in Equations 1314 was permitted to equal 0, time-invariant conditions" in Lines 5-6, Page C3 in AC2: Response to Reviewer 2. $\lambda$ was introduced later under Equation 16 but not used in Equation 13-14. Perhaps, an equation similar to Equation 16 is required to formulate u here. Also, it would be arbitrary that how the functional form for u was selected. Can it be justified?

**Response to R3C19**: The authors recognize that some of the terminology is confusing, and will rearrange the methodology for clarity (see Response to R3C16). There are 2 domains in each CV, fast and slow. The volumes of the domains are determined by equation 12. The estimated value of $\theta_o$ is used to estimate the volume of water in the slow domain while the different of $\theta$ and $\theta_o$ is used to estimate the volume of water in the fast domain. The authors will add in the linear definition of $u_F$ for clarity. A linear function was used for simplicity and to reduce parameterization. Within the function, if there is no slope the function has no time-variance, and permits the assessment of how sensitive the soil is to time-variant conditions.

**R3C20**: Model performance measure: Equation 19: Was the adjusted NSE newly developed in this study, or are there any references to cite?

**Response to R3C20**: The adjusted NSE is newly developed to better account for the large variability of measurement in the soil water. This metric was created throughout model development as it was determined that the mode efficiencies (not limited to NSE) were highly sensitive to the event in January 2016 (1 in 200 year event). Using the mean value on each day was not representative of the measurements, as each day presented different uncertainties of measurement.

**R3C21**: It is unclear what samples were used in the density estimation. Thus, I had to

guess that the replicated samples (n = 4) were used to construct the density. It is also not clear what bandwidth was used for the kernel density estimation.

**Response to R3C21**: The soil samples (n = 5) were used to derive the distribution. The manuscript will be revised to clarify how p in Eq. 19 was estimated. The use of kernel density was misstated. The kernel density estimations were only used for visual inspection of the results and a proxy of the GLUE method estimation, not within the NSE metric. Rather, the normal distribution was used (as the reviewer suggested in R3C22).

**R3C22**: Moreover, wouldn't the (perhaps chosen arbitrarily) bandwidth plays an important role in considering the measurement uncertainty in the adjusted NSE? I wonder what the benefit of using the kernel density would be compared to the likelihood functions which considers the measurement uncertainty in a statistically more rigorous way (by using statistics of the measurement such as standard deviation). I don't think the use of the kernel density would be a better way of accounting the uncertainty than using such likelihood functions.

**Response to R3C22**: As mentioned in Response to R3C21, it was misstated in the manuscript that kernel density estimation was used within the adjusted NSE estimation. The standard deviation of the samples was used with the normal distribution to estimate the probability (p) used in Eq. 19. Rather than fully addressing the uncertainty of the model, this method was used to reduce emphasis on sample days with high uncertainty. For example, the 1-in-200 year event (January 2016) was highly depleted from some samples, and less depleted for others. Most efficiency criteria are highly sensitive to this sample day as it has a larger deviation from the other sample days. However, there is high uncertainty associated with the sample day. Other evaluation tools to include measurement uncertainty (e.g. Kuppel et al., 2013), use a similar weighting method (e.g. covariance matrices) to reduce the influence of high uncertainty days on the overall model performance.

**R3C22**: It is not described well how the 100 parameter sets were chosen using the multiple NSEs (for different types of measurements and different measures). The only description is: "The "best" calibrations were selected using the $NSE_{adj}$ for all measurements (..) and a cumulative distribution function (Ala-Aho et al., 2017)" in Lines 2-4 on Page 9. It is unclear what the "cumulative distribution function" is in this context until one looks at the cited paper. More detailed description is required so that potential readers can grasp what the authors did without looking at the cited paper. (By the way, the description (Equations 6-8) in the cited paper is written with typos, so it was hard to understand the method. Thus, I think the equations should be re-written in this paper). Moreover, selecting the 100 best parameter sets (not 200, 1000, ..) is quite arbitrary, and the arbitrariness makes it difficult to interpret the model's uncertainty estimation.

**Response R3C22**: The authors direct the reviewer to the Initial Response to Reviewers 1 and 2 for how the 100 best parameter sets were selected. The parameter sets were based on minimum efficiency criteria rather than arbitrarily selected. The authors will include a more details explanation of how the parameters were selected, and subsequently ranked using the cumulative distribution function (CDF):

$$n(X) = (\cap_{i=1}^{5} \cap_{j=1}^{3} P_{i,j}(X \leq x))/100$$

where $P_{i,j}$ is a CDF for a model layer $i$ with efficiency criteria $j$, and $n(X)$ is the number of simulations meeting the objective ($X \leq x$).

**R3C23**: This part, mainly the discussion, was very hard to read as I don't have a clear picture of the developed model. Thus, I wrote only a few comments on this part at this stage. The authors stated that "this model structure does not make the assumption that uptake is time-variant or time-invariant" (for example, in Lines 6-7 on Page C3 in AC2: Response to Reviewer 2), and they argued that the model and data supported the time-variant hypothesis as perhaps the NSEs were higher when the time-variability was allowed (when the parameters $u_F$ is time-variant). I don't agree with the statements for two reasons. First, the criteria for choosing the time-invariant model is unclear. Second,

and more importantly, I don't think the authors have enough data to discuss the time-variability, and the detected time-variability could be an artifact. I will discuss these in more detail in the following. First, when can you say the beta distribution in Equations 14–15 was time-invariant? When the calibrated parameter (let's say $\lambda$) is "exactly" zero? Or, when the absolute value of $\lambda$ is less than a certain threshold? Moreover, it seems to me that the isotope dataset (presented in Figure 3) is perhaps not sufficient to test the time-variability. With the above (threshold) issue in mind, perhaps the best way to test the hypothesis on the time-variability is to see how the model works for two different cases: one with the $\lambda$ parameter set to 0 and another by allowing calibration of the parameter. If the authors can identify several periods when the model with $\lambda$ = 0 captures the observed time-variability, the authors perhaps can say that the time-variable model was required. However, I don't think the dataset is enough to be used for this, and perhaps the model still would do a relatively good job with the $\lambda$ parameter set to 0. Thus, I suggest the authors show the model results with the $\lambda$ parameter set to zero. As the use of NSE would not be sufficient to discuss it, the timeseries of model results (similar to Figure 3) should be included (at least in Supplemental material) so that readers can agree to the argument on the time-variability

**Response to R3C23**: The authors will include the parameter distributions in the appendix of the revisions as well as the sensitivity analysis. The parameter, $\lambda$, is one of the most sensitive parameters, which defines how variable the SAS function is in time. We did not previously include a direct comparison of time-variant vs time-invariant due to the observed values of calibrated $\lambda$. The values of $\lambda$ determined through calibration showed values trending away from values of 0, or even values near zero. Parameter values trending towards values of zero would indicate that the soils are more likely time-invariant than time-variant. We calibrated mean values of $\lambda$ of 3.7 for Site A and 2.1 for Site B. For improved comparison, we will provide a time-series of time-invariant and time-variant solutions in the appendix.

**R3C24**: It should be described better with an equation of how the SAS functions in

Figure 6 were estimated.

**Response to R3C24**: The equation will be included in the methods section.

**R3C25**: Comparison of the estimated range of water age. Page 20, Lines 24-25: It seems to me that the ranges overlapped each other quite a lot; thus, it is hard to agree with the statement.

**Response to R3C25**: While there are brief periods of overlap of the evaporation and root-uptake, the difference between the two fluxes is amplified in the spring and summer of 2016. During this period there are both low soil moisture (prior to June) and high soil moisture conditions (mid June). It is also during these periods that the evaporation and root-uptake are at the greatest values, and the differences between the two will be the largest. This distinct periods will be discussed further in the revised manuscript.

**References**

Benettin, P., Soulsby, C., Birkel, C., Tetzlaff, D., Botter, G. and Rinaldo, A.: Using SAS functions and high-resolution isotope data to unravel travel time distributions in headwater catchments, Water Resour. Res., 53, 1864-1878, doi:10.1002/2016WR020117, 2017.

Kuppel, S., Chevallier, F., and Peylin, P.: Quantifying the model structural error in carbon cycle data assimilation systems, Geosci. Model Dev., 6, 45-55, doi: 10.5194/gmd-6-45-2013, 2013.

Sprenger, M., Tetzlaff, D., Buttle, J., Laudon, H., Leistert, H., Mitchell, C., Snelgrove, J., Weiler, M. and Soulsby, C.: Measuring and modelling stable isotopes of mobile and bulk soil water, Vadose Zo. J., 2018.